# CryoEM structure of the SLFN14 endoribonuclease reveals insight into RNA binding and cleavage

Justin Van Riper [1,2,5], Arleth O. Martinez-Claros[1,2,5], Lie Wang[1], Hannah E. Schneiderman [3], Sweta Maheshwari[1,2,4] & Monica C. Pillon [1,2,3] ✉

The SLFN14 endoribonuclease is a post-transcriptional regulator that targets the ribosome and its associated RNA substrates for codon-bias translational repression. SLFN14 nuclease activity is linked to antiviral defense and platelet function. Despite its prominent role in gene regulation, the molecular signals regulating SLFN14 substrate recognition and catalytic activation remain unclear. SLFN14 dysregulation is linked to human diseases, including ribosomopathies and inherited thrombocytopenia, thus underscoring the importance of establishing the signals coordinating its RNA processing activity. Here, we reconstitute active full-length human SLFN14 and report a high-resolution cryoEM reconstruction of the SLFN14•RNA complex. The structure reveals a medallion-like architecture that shares structural homology with other SLFN family members. We unveil a C-terminal hydrophobic intermolecular interface that stabilizes the SLFN14 homodimer without the need for additional molecular signals. We describe compact sequence-independent RNA binding interfaces and highlight the environment of the SLFN14 disease hotspot at the RNA cleft entrance. We show that the SLFN14 endoribonuclease has broad site-specificity in the absence of modified native tRNA, a characteristic not shared with its SLFN11 family member. Finally, we demonstrate that metal-dependent acceptor stem cleavage requires the SLFN14 E-EhK motif and uncover its unexpected parallel with other virus-activatable nucleases.

The Schlafen (SLFN) family of host restriction factors are important gene regulator orchestrating viral defense, differentiation, immunity, and chemosensitivity[1–7]. SLFN are interferon-stimulated genes found throughout vertebrates, with gene duplications resulting in six human genes (*SLFN5, SLFN11, SLFN12, SLFN12L, SLFN13, SLFN14*)[8–10]. Mounting evidence indicates that upon their activation, SLFN proteins contribute to translational reprogramming through poorly understood RNA interactions[3,7,11]. SLFN11 is a pattern recognition receptor that binds intracellular single-stranded DNA triggering its translocation to the cytosol for type-II transfer RNA (tRNA) cleavage and subsequent innate immune signaling and apoptosis[7,12–14]. SLFN12 assembly with phosphodiesterase 3A (PDE3A) is licensed by its association with Velcrin compounds, stimulating tRNA nuclease activity for ribosome stalling and cell death[15]. How the remaining SLFN post-transcriptional regulators are activated for RNA processing is unknown. Moreover, how SLFN-directed tRNA cleavage results in translational regulation is poorly understood. Considering the SLFN family lies at the nexus of virus-host interactions, there is a pressing need to understand the

[1]Verna and Marrs McLean Department of Biochemistry and Molecular Pharmacology, Baylor College of Medicine, Houston, TX, USA. [2]Therapeutic Innovation Center (THINC), Baylor College of Medicine, Houston, TX, USA. [3]Department of Structural Biology, Jacobs School of Medicine and Biomedical Sciences, University at Buffalo, SUNY, Buffalo, NY, USA. [4]Present address: Merck & Co., Inc., Rahway, NJ, USA. [5]These authors contributed equally: Justin Van Riper, Arleth O. Martinez-Claros. ✉e-mail: mpillon@buffalo.edu

structural and molecular underpinnings governing their RNA processing activities for translational control.

The SLFN family is an emerging class of RNA-targeting endonucleases. An early report demonstrates SLFN14 ribonuclease activity[16]. Rabbit SLFN14 associates with stalled ribosomal elongation complexes and catalyzes the cleavage of transfer, ribosomal, and messenger RNAs in vitro[16]. While its physiological targets and precise mechanism of action are unknown, SLFN14 localizes to the cytosol, supporting the hypothesis that SLFN14 may process the translation machinery or its associated RNA substrates[17,18]. Correspondingly, SLFN14 ribonuclease activity is required for translational repression of immunodeficiency virus 1 (HIV-1) protein synthesis, and, like SLFN11 and SLFN12, repression is selective towards transcripts enriched for rare codons[18–20]. SLFN11, SLFN12, and SLFN13 are also bona fide ribonucleases and share a common processing site at the base of the tRNA acceptor stem[15,21–24]. SLFN11 cleaves all leucine tRNAs (tRNA$^{Leu}$), several serine tRNAs (tRNA$^{Ser}$), and the initiator methionine tRNA (tRNA$^{iMet}$)[13]. Meanwhile, SLFN12 displays strict cleavage selectivity for tRNA$^{Leu}$ harboring the TAA anticodon[15]. It is still unclear whether SLFN12L has catalytic activity and RNA targeting preference. Moreover, SLFN5 does not harbor nuclease activity and binds DNA for transcriptional control[24,25]. While substrate preference is starting to come to light for this important family of gene regulators, how SLFN proteins recognize their physiological targets remains unclear.

Structures of SLFN family members unveil a common core topology. SLFN members are multidomain proteins comprising an N-terminal domain (NTD) often associated with RNA cleavage, a middle domain (MD), and, for select members, a C-terminal domain (CTD) (Fig. 1a and Supplementary Fig. 1)[21–24,26]. The NTD contains topologically equivalent N-terminal (N) and C-terminal (C) lobes juxtaposed by a small bridging subdomain[23,26]. The resulting U-shaped valley forms a concave RNA cleft capable of accommodating nucleic acid[21,23,24]. The C-lobe harbors characteristic elements of the SLFN family, including a

conserved glutamate residue essential for RNA cleavage and a putative phosphorylation site that has been validated in SLFN11 to modulate RNA cleavage activity (Fig. 1a)[27,28]. The MD has a mixed α/β fold linking the RNA cleft to the CTD, which shares structural homology with superfamily I (SFI) DNA/RNA helicases[22]. SLFN11 and SLFN12 self-associate whereas SLFN13 and SLFN5 are monomeric in solution[21–24]. To date, there are no structures of SLFN14 or SLFN12L, and there is limited structural information describing how SLFN proteins recognize their RNA targets. Recent structures of SLFN11 bound to tRNA reveal a single tRNA molecule bound to the homodimeric RNA cleft[22,28]. These asymmetric SLFN11 structures highlight multiple contacts within the acceptor stem, T-arm, and variable arm of tRNA[28]. Whether other SLFN family members adopt a similar SLFN11-like tRNA binding mode is unclear. Moving forward, a complete repertoire of high-resolution SLFN structures illustrating RNA-bound states will be necessary to define universal and specialized features of this important RNA processing family.

A distinguishing feature of SLFN14 is its hematopoietic-specific expression and role in platelet function. In the absence of stress, SLFN14 is exclusively expressed in hematopoietic cells[29,30]. While its precise molecular contributions remain undefined, SLFN14 expression and stability are essential for proplatelet formation, platelet morphology, and platelet activation[30–34]. SLFN14 dysregulation is causative for inherited thrombocytopenia, a collection of disorders typified by low platelet count, excessive bleeding, and ribosome dysfunction[30,33]. Interestingly, the phenotype of SLFN14 dysregulation is species-specific, with mice displaying normal thrombopoiesis but deficient erythropoiesis[29]. Patient genotyping has uncovered an SLFN14 hotspot comprising missense mutations K218E, K219E/N, V220D, and R223W[31,32,34]. These SLFN14 disease variants do not result in a ribosomal RNA (rRNA) cleavage defect[17,30], but pleiotropic effects, including the upregulation of ribosomal genes and rRNA processing pathways[30]. While SLFN14 is an important post-transcriptional regulator, the

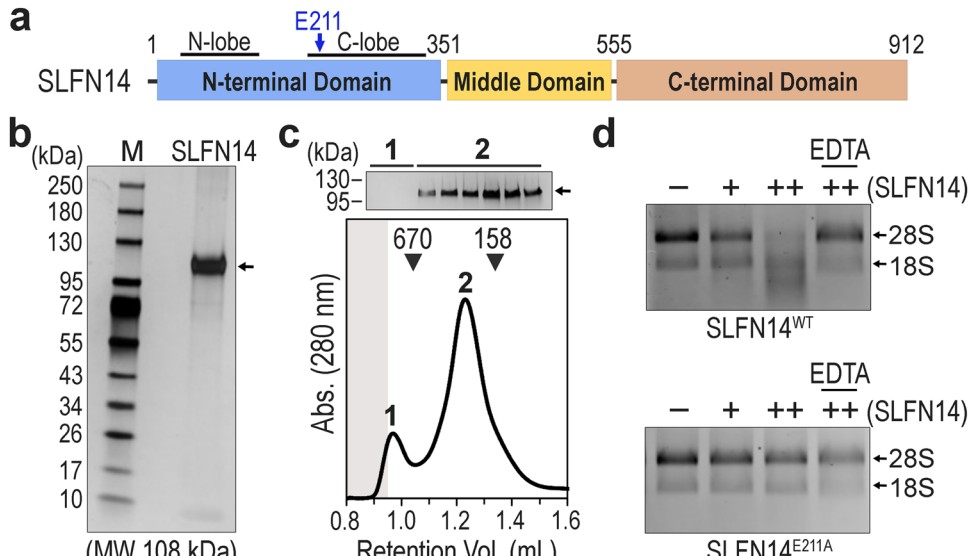

**Fig. 1 | Reconstitution of the human SLFN14 ribonuclease. a** Cartoon diagram of human SLFN14 with the numbering defining the amino acid domain boundaries. The SLFN14 N-terminal domain is shown in blue, the middle domain is colored yellow, and the C-terminal domain is depicted as brown. Black bars define the N-lobe and C-lobe subdomains. The dark blue arrow marks the location of the SLFN14 E211 catalytic residue. **b** SDS-PAGE analysis of purified recombinant human SLFN14. Gel was visualized by silver stain, and the theoretical molecular weight of SLFN14 is shown in brackets. Three experiments were repeated independently with similar results. **c** Size-exclusion chromatography of recombinant SLFN14. Molecular weight (kDa) standards are indicated at the top of the plot. The shaded region

demarcates the void volume of the chromatography column. SDS-PAGE analysis of peaks 1 and 2 are shown above and visualized by silver stain. Two experiments were repeated independently with similar results. **d** Representative in vitro RNA cleavage assay of SLFN14 wild-type (WT) and catalytic (E211A) variants incubated with isolated total human RNA (1 μg) and Mg$^{2+}$ metal ion in reaction buffer supplemented with excess EDTA, where indicated. – marks reactions set in the absence of SLFN14 variants, whereas + (0.5 μM) and ++ (2 μM) identify reactions with SLFN14 variants. Mammalian 28S and 18S rRNA species are identified by black arrows. Three experiments were repeated independently with similar results. Source data are provided as a Source Data file.

molecular details describing its precise role in megakaryocyte differentiation and platelet production remain poorly understood.

In this study, we establish a recombinant protein production workflow to isolate full-length human SLFN14 and characterize its structure and ribonuclease activity in vitro. We report a cryoEM structure of SLFN14 to an overall resolution of 2.73 Å. We unveil the architecture of the constitutive SLFN14 homodimer bound to native RNA (SLFN14•RNA). The SLFN14•RNA structure reveals a shared architecture with other SLFN family members and RNA binding interfaces along the N- and C-lobes of the RNA cleft. The structure uncovers the structural environment of the SLFN14-related thrombocytopenia hotspot mutations, suggesting dysregulation of RNA binding within the C-lobes. SLFN14 cleaves RNAs in a metal-dependent manner and often requires native tRNA for efficient on-target cleavage of the acceptor stem. We analyzed SLFN14 mutations in the catalytic C-lobe to gain insight into the importance of this region. We describe the SLFN14 residues essential for catalysis and demonstrate its analogy to the D-(D/E)hK catalytic motif of other viral-activated prokaryotic nucleases.

## Results

### Reconstitution of the full-length SLFN14 ribonuclease

We established a workflow to produce full-length human SLFN14 ribonuclease. C-terminal truncated forms of SLFN14 retain nucleolytic activity; however, full-length SLFN14 has yet to be reconstituted and characterized due to the insolubility of its CTD[16,17]. We developed a reconstitution protocol to generate full-length recombinant human SLFN14 (residues 1–912) from HEK293 cells, which lack detectable endogenous SLFN14 at the mRNA and protein levels[18,19]. We transiently transfected HEK293 cells with plasmid DNA encoding C-terminally Flag-tagged SLFN14 (herein SLFN14) and isolated the recombinant protein by anti-Flag affinity chromatography followed by size exclusion chromatography. SDS-PAGE analysis of purified SLFN14 reveals a single protein band with an apparent molecular weight of ~100 kDa, corresponding well with the expected molecular weight of SLFN14 (108 kDa) (Fig. 1b). The identity of the isolated band was confirmed to be human SLFN14 by mass spectrometry. To determine the stability of SLFN14, we performed size exclusion chromatography at a physiological salt concentration using a calibrated analytical column. SLFN14 resolves as a single homogenous peak suggesting SLFN14 is stable in solution (Fig. 1c). Mass photometry reveals a prominent salt-resistant SLFN14 homodimer and a minor salt-sensitive homotetramer population (Supplementary Fig. 2). In the absence of a regulatory signal, the stable SLFN14 dimer is distinct from other SLFN family members, which have been reported as either monomers (SLFN5, SLFN13) or transient homodimers (SLFN11, SLFN12)[21–24]. These data reveal that the mammalian expression host is suitable for producing soluble full-length SLFN14 protein.

Next, we confirmed that recombinant SLFN14 retains ribonuclease activity. To verify RNA endonucleolytic activity, we purified native 28S and 18S rRNA from HEK293 cells. The purified rRNA was incubated with SLFN14 protein in the presence of $Mg^{2+}$ metal ion, and the RNA products were resolved on denaturing formaldehyde gels and visualized by ethidium bromide. As previously reported[16], ribosomal RNA cleavage correlates with increasing concentration of SLFN14 protein (Fig. 1d). Since the SLFN11 structure shows the catalytic C-lobe coordinates a manganese metal ion[22,28], we next sought to determine whether SLFN14 is a metal-dependent nuclease. Reaction mixtures containing magnesium and manganese ions support comparable levels of SLFN14 rRNA cleavage activity (Supplementary Fig. 3). The cleavage reaction was repeated in the presence of the metal chelating agent EDTA. The cleavage of rRNA is suppressed upon the addition of EDTA, confirming the presence of a metal-dependent ribonuclease (Fig. 1d). To verify our purified SLFN14 sample is not contaminated with a mammalian ribonuclease, we assessed rRNA cleavage using an established SLFN14 catalytic variant harboring a missense mutation to an invariant C-lobe residue (Fig. 1a)[16]. The purified SLFN14 E211A catalytic variant does not cleave the rRNA substrates (Fig. 1d), confirming full-length human SLFN14 is a functional metal-dependent ribonuclease.

### SLFN14 cleaves synthetic tRNA outside of the acceptor stem

We characterized SLFN14 cleavage activity and found it processed synthetic tRNA differently than SLFN11. To compare the RNA cleavage properties of SLFN14 with a well-characterized family member, we produced full-length recombinant SLFN11 (residues 1–901) and validated the sample by SDS-PAGE analysis and mass spectrometry (Supplementary Fig. 4a). We carried out an independent protein titration series of SLFN14 and SLFN11 with in vitro transcribed (synthetic) tRNA$^{Ser}$ harboring the TGA anticodon in the presence of $Mg^{2+}$ metal ion (Fig. 2a). RNA cleavage products were analyzed by urea denaturing gels and visualized by SYBR gold. Despite an earlier report that SLFN11 is solely active on tRNA when $Mn^{2+}$ ion is added to $Mg^{2+}$ containing reaction buffer, we observe that $Mg^{2+}$ alone is sufficient to support SLFN11 and SLFN14 ribonuclease activity (Fig. 2a)[22]. However, supplementing the tRNA cleavage reaction with $Mn^{2+}$ metal enhances SLFN cleavage activity in vitro, resulting in additional RNA fragments (Supplementary Fig. 4b, c). Like SLFN14, an SLFN11 C-lobe catalytic variant (E214A) does not cleave RNA, attributing the observed tRNA processing pattern to recombinant SLFN11 protein (Supplementary Fig. 4d). SLFN11 cleaves synthetic tRNA$^{Ser}$ more efficiently than SLFN14, suggesting a possible distinction in substrate preference. SLFN11 produces a single prominent RNA cleavage product above the 70-nt marker (Fig. 2a). This correlates well with the previous report that SLFN11 cleaves the synthetic tRNA$^{Ser}$ acceptor stem, generating a visible 75-nt 5′-fragment and a corresponding 10-nt 3′-fragment that is difficult to detect using conventional methods[22]. On the other hand, SLFN14 cleaves synthetic tRNA$^{Ser}$ at multiple sites, generating four distinct fragments of approximately 75-, 60-, 40-, and 30-nt in length (Fig. 2a). This alternate cleavage pattern demonstrates an intrinsic difference of in vitro tRNA processing between SLFN14 and the SLFN11 family member.

We analyzed the synthetic tRNA products to identify the location of the SLFN cleavage events. SLFN nucleases catalyze a metal-dependent hydrolysis reaction resulting in 3′-hydroxyl and 5′-monophosphate RNA ends[28,35]. To identify SLFN-derived 3′-fragments, we incubated the reaction mixtures with XRN-1, an exoribonuclease requiring a 5′-monophosphate for RNA degradation[36]. As expected, the unprocessed synthetic tRNA$^{Ser}$ substrate lacks a 5′-monophosphate and is resistant to XRN-1 RNA decay (Fig. 2b). Likewise, the SLFN11-derived 75-nt fragment is resistant to XRN-1, confirming it is a 5′-fragment and likely the canonical SLFN11 acceptor stem cleavage product reported previously (Fig. 2b)[22]. The SLFN14 60-nt and 40-nt RNA products are sensitive to XRN-1, indicating they are 3′-fragments, and the 75-nt and 30-nt products are resistant to XRN-1, marking them as 5′-fragments (Fig. 2b). Two primary endonucleolytic cleavage events could explain this pattern. Cleavage at the canonical acceptor stem site could result in the 75-nt 5′-fragment, whereas cleavage of the anticodon arm could result in the 30-nt 5′-fragment and its corresponding 60-nt 3′-fragment. The 40-nt 3′-fragment could result from a secondary cleavage event within the nicked tRNA. Intriguingly, the appearance of the 40-nt 3′-fragment correlates with the depletion of the 75-nt 5′-fragment. This suggests that following acceptor stem cleavage, SLFN14 may further process the nicked synthetic tRNA within the anticodon arm. Taken together, SLFN11 strictly processes the acceptor stem of synthetic tRNA$^{Ser}$, while SLFN14 displays broader site-specificity in the conditions tested.

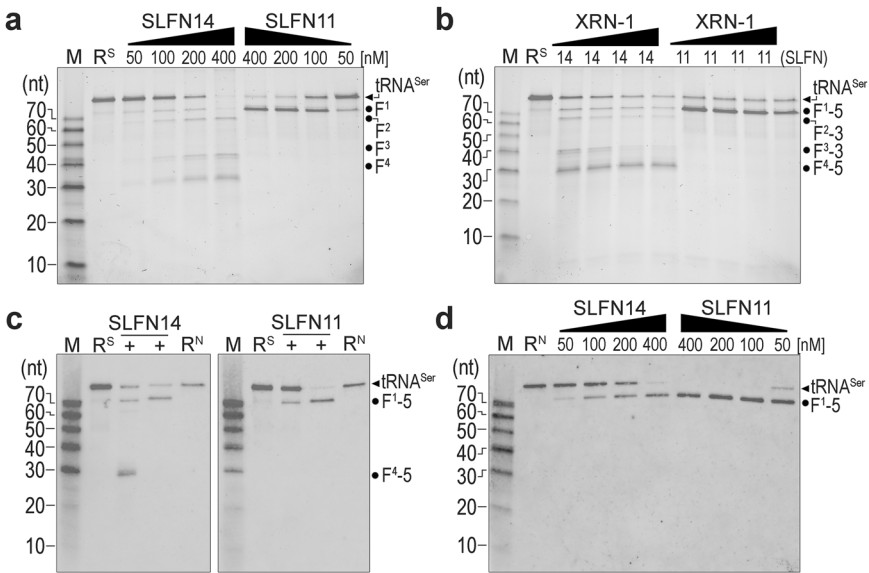

**Fig. 2 | Cleavage of synthetic and native tRNA^Ser by SLFN14 and SLFN11.**
**a** Representative urea-PAGE gel of synthetic tRNA^Ser (100 nM) cleavage by SLFN14 and SLFN11 protein titrations (black triangles, 50, 100, 200, and 400 nM). Synthetic RNA mixture without protein (R^S) is a negative control. Gel visualized by SYBR Gold. **b** Representative denaturing gel of synthetic tRNA^Ser (100 nM) incubated with SLFN protein (200 nM) and a yeast XRN-1 exoribonuclease titration (black triangles defining 16, 8, 4, and 2 × 10^-4 units). Gel is visualized by SYBR Gold. XRN-1 sensitive 3'-fragments are denoted by the number three, and XRN-1 resistant 5'-fragments are labeled with the number five. **c** Representative Northern blot of synthetic tRNA^Ser

(R^S, 100 nM) and native tRNA^Ser (R^N, 100 nM total small RNA) processing by SLFN14 and SLFN11 ( + , 200 nM). **d** Representative Northern blot of native tRNA^Ser (100 nM total small RNA) in the presence of SLFN14 and SLFN11 protein titrations (black triangles, 50, 100, 200, and 400 nM). M defines the 70-10-nt RNA ladder with its corresponding nucleotide size on the left. Black arrowheads mark unprocessed tRNA^Ser and black dots define cleaved tRNA^Ser products. F defines a fragment. For all representative blots presented in this figure, three experiments were repeated independently with similar results. Source data are provided as a Source Data file.

## SLFN14 requires native tRNA for acceptor stem selectivity

We next sought to determine whether intrinsic features of native RNA influence SLFN14 tRNA processing. We isolated short cellular-sourced RNA transcripts (< 200 nucleotides) from HEK293 cells to enrich for native tRNAs that presumably retain their post-transcription modifications[37]. We compared SLFN14 processing with unmodified synthetic and modified native tRNA^Ser by Northern blot analysis using a probe targeting the 5'-end of tRNA^Ser. As described above, cleavage of synthetic tRNA^Ser by SLFN14 results in ∽75-nt and 30-nt 5'-fragments, confirming our earlier interpretation that SLFN14 cleaves both the acceptor stem and anticodon arm (Fig. 2c). Yet, intriguingly, SLFN14 processing of native tRNA^Ser exclusively produces the ∽75-nt fragment, highlighting a change in its selectivity, where it exclusively targets the acceptor stem site (Fig. 2c). To investigate whether RNA modifications encoded within the native RNA could explain its altered site-specificity, we produced synthetic and native 5S rRNA, a substrate reported to be unmodified in cells[38]. SLFN14 cleaves both unmodified synthetic and native 5S rRNA, producing indistinguishable RNA products regardless of their source (Supplementary Fig. 5). While SLFN14 appears to exhibit a dependency on RNA modification(s), SLFN11 site-specificity is insensitive to tRNA^Ser modifications, cleaving both unmodified synthetic and modified native substrate at the acceptor stem (Fig. 2c). In addition to the observation that SLFN14 is substantially less efficient at cleaving native tRNA^Ser than SLFN11 (Fig. 2d), these data highlight differences in SLFN14's site-specificity and catalytic efficiency that may be important distinguishing features of the SLFN family.

Next, we investigated whether SLFN14's site-specificity is altered with other native tRNA substrates. We measured SLFN14 RNA cleavage in vitro using type-I tRNAs, tRNA^iMet and lysine tRNA (tRNA^Lys). SLFN14 inefficiently cleaves both unmodified synthetic tRNA^iMet and tRNA^Lys outside of the acceptor stem, demonstrating spurious RNA processing activity (Supplementary Fig. 6). Conversely, in the presence of modified native tRNA, SLFN14 exhibits site-selective, albeit relatively weak,

nuclease activity towards the acceptor stem (Supplementary Fig. 6). SLFN11 is a more productive tRNA nuclease, efficiently depleting intact native tRNA^iMet and tRNA^Lys through robust acceptor stem cleavage (Supplementary Fig. 6). Second, we monitored SLFN14 processing using a type-II tRNA substrate, tRNA^Leu, to determine whether modification-dependent site-specificity is a universal property of SLFN14 cleavage. SLFN14 exclusively cleaves the acceptor stem regardless of the tRNA^Leu source, implying that while modifications often influence site-specificity in vitro, they are not a universal determinant of SLFN14 processing. Correspondingly, SLFN14 also cleaves unmodified synthetic and modified native selenocysteine tRNA (tRNA^SeC) similarly, producing indistinguishable RNA products (Supplementary Fig. 6). However, SLFN14 cleaves tRNA^SeC outside of the acceptor stem, which we attribute to its atypical tRNA fold, including an elongated acceptor stem[39]. SLFN11 does not cleave synthetic or native tRNA^SeC, suggesting a more strict selectivity mechanism compared to its SLFN14 counterpart. Overall, this work implies that SLFN14 often, but not always, relies on modified native tRNAs for acceptor stem site-specificity.

## Architecture of the dimeric SLFN14 ribonuclease

To gain structural insight into the organization of the SLFN14 ribonuclease, we solved the cryoEM structure of enzymatically active human SLFN14 bound to native RNA (SLFN14•RNA). Initially, we vitrified the SLFN14 sample in the absence and presence of native tRNA purified from brewer's yeast and collected independent single-particle cryoEM datasets. Image processing of the dataset collected without native tRNA revealed a strong bias for side views of SLFN14 particles, whereas 2D classification with native tRNA produced a bias for front views. The standard approach of supplementing the protein sample with detergents to alter the properties of the air-water interface did not improve the distribution of particle orientation. To ensure we observed sufficient views of the sample, we collected a third cryoEM dataset in the presence of native tRNA using a 25° stage tilt. We combined all three

datasets for subsequent image processing to reveal two prominent asymmetric SLFN14 reconstructions, state 1 and state 2, at a resolution of 2.73 Å and 3.11 Å, respectively (Supplementary Figs. 7, 8). Overall, the architecture between states 1 and 2 are similar, with modest differences at the periphery of the SLFN14 particles (Supplementary Fig. 9). The quality of the state 2 reconstruction was insufficient for model building, however, state 1 revealed well-resolved density for the NTD, MD, and CTD of an SLFN14 homodimer (Fig. 3a, b and Supplementary Figs. 8 and 10). We observed additional density for $Mg^{2+}$ metal ions within the catalytic C-lobes and $Zn^{2+}$ metal ions within the conserved zinc fingers (Supplementary Fig. 10). The electron scattering map reveals a medallion-shaped particle with dimensions of 110 x 65 x 55 Å (Fig. 3a). Protomer A closely resembles that of protomer B (root mean square deviation (r.m.s.d.) of 0.652 Å), with a slight conformational change in the position of the C-lobe (Supplementary Fig. 11). The SLFN14 homodimer is supported through two intermolecular contacts, an NTD-NTD' dimerization interface and a CTD-CTD' dimerization interface. The overall SLFN14 architecture is reminiscent of tRNA-bound SLFN11 homodimer structures[28]. Together, the SLFN14 cryoEM reconstruction unveils a canonical SLFN dimeric architecture composed of a head-to-head, tail-to-tail structural arrangement.

Visual inspection of the SLFN14 reconstruction reveals additional density within the RNA cleft. The unassigned density cannot be described by unmodeled SLFN14 protein loops, indicating a bound substrate. The shape of the unassigned density resembles the structure formed by double-stranded nucleic acid, and the width corresponds to that of a Watson-Crick base pair. Despite adding native tRNA to the cryoEM sample, the density is insufficient to model an intact tRNA molecule. We suspect that co-purified cellular-sourced nucleic acid, low substrate occupancy, substrate heterogeneity, and an active SLFN14 ribonuclease limit the signal and resolution of the unassigned density. Consistent with the presence of cellular-sourced nucleic acid, the calculated molecular mass of SLFN14 ($232 \pm 5.89$ kDa) is consistently larger than its theoretical dimer (215 kDa), indicating an additional component to the sample (Supplementary Fig. 2). When the protein sample is exposed to high salt conditions, the calculated mass is smaller and aligns well with the theoretical mass of apo dimeric SLFN14 ($212 \pm 3.21$ kDa), suggesting the unidentified component was released (Supplementary Fig. 2). While we could not determine the identity of this component, the strong absorbance of the SLFN14 sample at 260 nm suggests it retains cellular nucleic acid during protein purification. Considering SLFN14 is a known RNA-binding protein[16], we modeled the unassigned density as RNA. A 3'-overhang duplexed RNA molecule was modeled into the asymmetric RNA cleft. The unassigned density lacks nucleobase features and was modeled using RNA of arbitrary sequence. The model RNA features three consecutive base pairs followed by two unpaired nucleotides, and two overhang nucleotides extending towards the catalytic C-lobe of protomer A. Additional density is present along both SLFN14 C-lobes; however, the lack of discrete features prevents modeling in this region of the reconstruction (Fig. 3a and Supplementary Fig. 10).

## CTD stabilizes the constitutive SLFN14 homodimer

The SLFN14 dimer adopts a head-to-head, tail-to-tail arrangement stabilized through the CTD molecular scaffold. Like the SLFN11 homodimeric architecture[28], the SLFN14 N- and C-termini self-associate to form NTD-NTD' and CTD-CTD' dimerization interfaces, respectively (Supplementary Movie 1). While the total surface area of the dimerization interfaces is similar between SLFN14 and SLFN11, the SLFN14 interface is notably more hydrophobic[40]. To determine whether the CTD is required for a stable SLFN14 dimer, we expressed an SLFN14 C-terminal truncation that lacks the CTD (SLFN14$^{\Delta CTD}$) and monitored its dimerization in solution by size exclusion chromatography. The SLFN14$^{\Delta CTD}$ variant elutes as two distinct peaks with a calculated molecular weight corresponding to a homodimer (160 kDa)

and monomer (69 kDa) (Supplementary Fig. 12). To confirm the necessity of the CTD for homo-oligomerization, we directly imaged SLFN14$^{\Delta CTD}$ particles by negative stain electron microscopy (Supplementary Fig. 12). 2D classification reveals distinct projections corresponding well with a monomer-dimer equilibrium. Likewise, the isolated SLFN14 NTD does not form a stable NTD-NTD' homodimer in solution, whereas the CTD is primarily dimeric (Supplementary Fig. 12). Therefore, unlike SLFN11, which forms a weakly associated homodimer in solution[22], SLFN14 adopts a stable homodimer.

The SLFN CTD shares structural homology with SFI RNA/DNA helicases, raising the question as to whether any SLFN members harbor nucleic acid remodeling activity. The SLFN14 homodimer contains sterically occluded Walker A and Walker B motifs, which are traditionally essential for ATP-dependent helicase activity (Supplementary Fig. 13a)[8,41–43]. To verify whether the SLFN14 CTD could support helicase function, we measured ATP binding and hydrolysis in vitro. In line with earlier reports that dimeric SLFN members are dormant ATPases[24,28], we do not detect a significant change in the SLFN14 melting temperature in the absence and presence of excess nucleotides (ATP, ADP, ATPγS) (Supplementary Fig. 13b). Moreover, we do not observe stimulation of its ATPase activity in the presence of nucleic acid as reported with well-known eukaryotic RNA helicases (Supplementary Fig. 13c)[44]. This analysis suggests that in the absence of a signal, the SLFN14 CTD is likely a dormant ATPase.

## Substrate binding at the SLFN14 RNA cleft

How SLFN proteins recognize their RNA substrates is a subject of intense investigation. Like SLFN11 and SLFN12[21,22,28], the dimeric SLFN14 NTD forms a U-shaped cleft that spans approximately 20 Å (Fig. 3b). The entrance of the RNA cleft is defined by large electropositive surfaces along the N- and C-lobes and is wide enough to accommodate double-stranded RNA. 3D classification reveals two distinct SLFN14•RNA conformational states (states 1 and 2) that are largely indistinguishable, except for a subtle wobble along the N- and C-terminal dimerization interfaces (Supplementary Movie 2). Specifically, the malleability of the RNA cleft is due to the unfixed C-lobes that tilt toward the center of the RNA cleft (Supplementary Figs. 9 and 11). The dynamics of the C-lobes facilitate the flexibility of its RNA binding site and may be important for binding different RNA substrates. Four conserved residues (H282, C284, C318, C319) also cluster at the base of the C-lobe to form a zinc finger for $Zn^{2+}$ metal ion coordination (Fig. 3b). Characterization of SLFN14 variants harboring missense mutations in its zinc finger reveals partial protein misfolding and the co-purification of mammalian chaperone machinery (Supplementary Fig. 14)[45]. This confirms that the SLFN14 zinc finger is an important structural element of the conformationally pliable RNA cleft.

The SLFN14•RNA structure reveals sequence-independent RNA binding surfaces within the RNA cleft. In our 3D reconstruction, the NTD of protomer A is better resolved than that of protomer B, indicating conformational heterogeneity across the RNA cleft (Supplementary Fig. 10). Each SLFN14 protomer interacts with the duplexed region of the RNA molecule forming compact RNA binding surfaces of ∼400 Å² along each N-lobe (Fig. 3c). Juxtaposed N-lobe α-helices display ordered polar and basic residues (e.g., K38, Q78, T82) along the sugar-phosphate backbone of the opposing strands, potentially counterbalancing the negative charge of the RNA molecule (Fig. 3c and Supplementary Movie 3). The charge at position R39 is often conserved and ordered in protomer B, where it is located near a phosphate group (N3') at the RNA cleft entrance, while it is disordered in protomer A. Conserved C-lobe residue S137 (protomer B) lies adjacent to a ribose moiety, raising the question as to whether it can interact with the 2'-hydroxyl to discriminate RNA (Fig. 3c, d). Highly conserved K213 (protomer A) is positioned next to two magnesium ions within the SLFN14 catalytic site and is sufficiently close to interact with the phosphate group of the terminal RNA

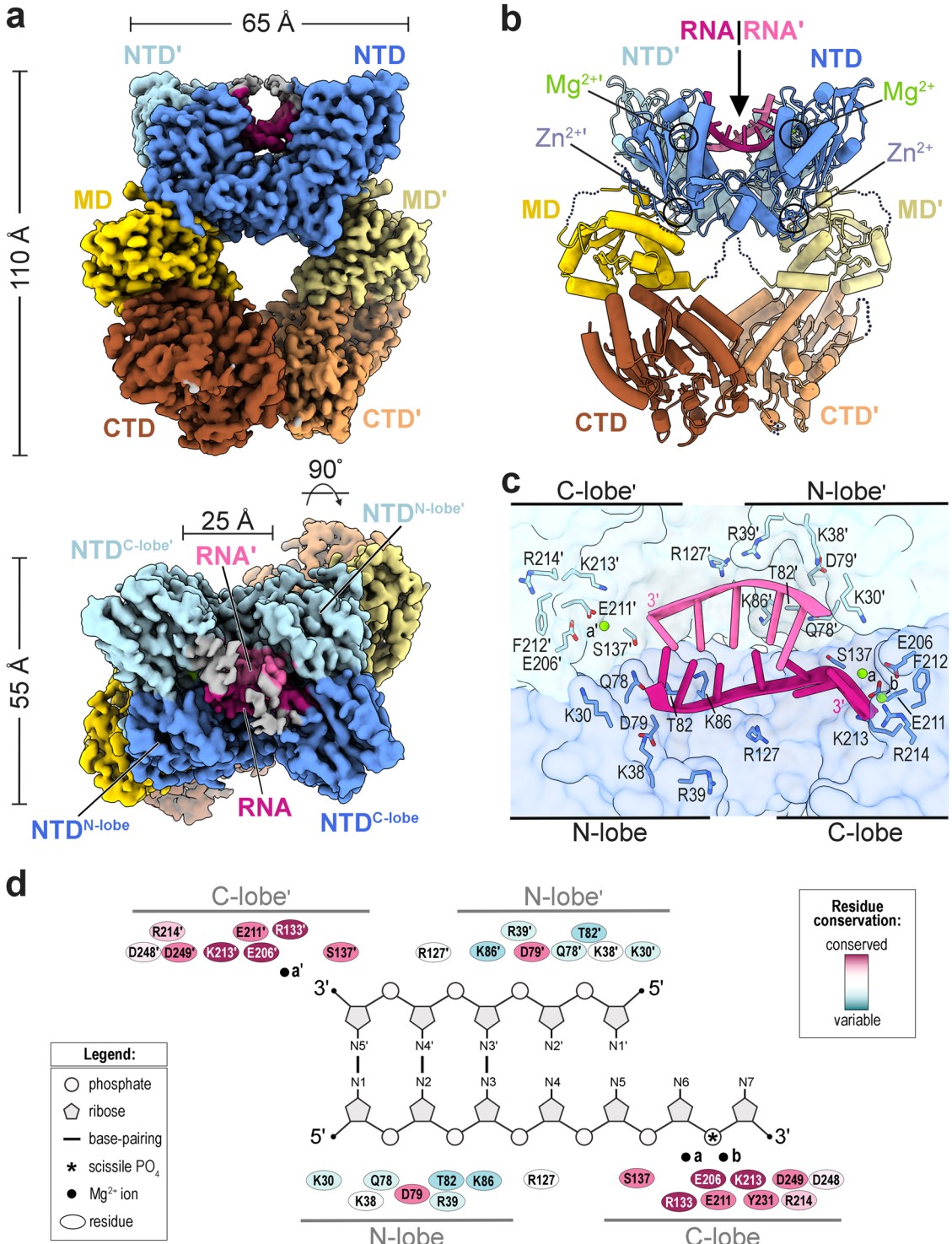

**Fig. 3 | Architecture of the RNA-bound SLFN14 complex. a** Orthogonal views of state 1 SLFN14•RNA cryoEM reconstruction shown in blue (NTD), yellow (MD), and brown (CTD). Density within the RNA cleft is shown in pink (RNA) and gray (unassigned). The dimeric RNA-bound SLFN14 protomers are distinguished by dark and light colors as well as the absence and presence of a prime symbol. General dimensions of the SLFN14 reconstruction are reported in Å. **b** Model of dimeric SLFN14•RNA colored as seen in (panel **a**). $Mg^{2+}$ and $Zn^{2+}$ metal ion sites are circled. **c** Inset of the SLFN14•RNA cleft. SLFN14 surface representation and cartoon of select N-lobe and C-lobe residues along the RNA interfaces. **d** 2D diagram of the SLFN14 N- and C-lobe RNA interfaces, where residues are colored based on Consurf sequence conservation[83].

nucleotide (N7), an interaction reminiscent to the conserved configuration of a scissile phosphate[46]. Collectively, electrostatic interactions within the RNA cleft provide a structural framework to understand how RNA molecules are bound by the SLFN14 ribonuclease.

To extend our investigation into SLFN14 tRNA binding, we modeled the tRNA^Ser-bound SLFN14 complex. We leveraged AlphaFold3 to generate an SLFN14-tRNA^Ser model with confidence metrics characteristic of a high-quality prediction[47]. We superimposed the prediction model with the SLFN14•RNA structure and calculated an r.m.s.d. of

1.3 Å (Fig. 4a). Overall, the SLFN14 protein fold is similar between our experimental data and the prediction model. The L-shape tRNA[Ser] molecule is predicted to be in an upright position with the T-arm and acceptor stem running across the base of the SLFN14 RNA cleft (Fig. 4a). Furthermore, the paired nucleotides of the T-arm are predicted to run along the juxtaposed SLFN14 N-lobes. The phosphodiester backbone of the predicted T-arm overlays with the duplexed region of the RNA observed in our SLFN14•RNA structure, suggesting the N-lobes may interact with the T-arm for tRNA binding (Fig. 4b, c). Strikingly, the predicted configuration of tRNA[Ser] aligns the canonical SLFN acceptor stem's scissile phosphate with the phosphate group of the 3′-terminal nucleotide, adjacent to two coordinated Mg[2+] ions in the C-lobe of protomer A in our cryoEM structure (Fig. 4c, d). Although this structural arrangement could, in principle, represent a scissile phosphate positioned for RNA hydrolysis or a post-cleavage product, the cryoEM reconstruction within this region is not of sufficient quality to conclusively determine the RNA cleavage state. Despite these limitations, the structural organization of the SLFN14 active site observed in protomer A of our cryoEM reconstruction is supported by recent tRNA-bound SLFN11 pre- and post-cleavage structures, demonstrating that the N-lobes directly associate with the T-arm of type-I and type-II tRNAs, positioning the acceptor stem at the neighboring catalytic C-lobe (Supplementary Fig. 15)[28].

Our cryoEM structure unveils the SLFN14-related thrombocytopenia hotspot at the entrance of the RNA cleft. Encoded within the C-lobe, the cluster of disease mutants K218E, K219E/N, V220D, and R223W are poised above the catalytic Mg[2+] metal ion site (Fig. 4d). While the cryoEM reconstruction describing the side chains of basic residues K218, K219, and R223 are poorly defined, the well-resolved main chain places them at the transition point between an amphipathic α-helix and a partially ordered loop. Together, these basic residues contribute to a discrete surface at the entrance of the RNA cleft that may assist in substrate recruitment (Fig. 4e). Consistent with a role in substrate binding, the equivalent C-lobe surface in tRNA-bound SLFN11 structures contains basic residues that form tRNA interfaces with the acceptor stem, T-arm, and variable arm[28]. Our structure also reveals the placement of the hydrophobic V220 residue on the opposite face of the corresponding amphipathic α-helix, contributing to a hydrophobic core. We suspect the pathological V220D mutation drives local structural distortions that destabilize the nearby RNA binding surface, which includes K218, K219, and R223. Together, comparisons with existing tRNA-bound SLFN11 structures propose an RNA recruitment or binding defect for the SLFN14 disease hotspot variants.

## Characterization of the SLFN14 ribonuclease active site

The SLFN14•RNA structure provides insight into RNA hydrolysis at its catalytic center. The SLFN14 C-lobe shares a conserved structural fold with its SLFN family members. This subdomain harbors a β-sheet that lines the RNA cleft and harbors the consensus ExxxxExK (where x is any residue) motif associated with SLFN catalytic function (Fig. 5a)[8,21–23,26]. This discrete SLFN element harbors two invariant glutamates (E206 and E211 in SLFN14), shown to coordinate two Mg[2+] cations in protomer A, and a nearby conserved lysine residue (K213 in SLFN14) that interacts with a nearby RNA phosphate group (Fig. 5b)[22]. In addition to the core ExxxxExK motif, SLFN members also encode a nearby invariant arginine (R133 in SLFN14), and two tandem aspartates (D248 and D249 in SLFN14) conserved in a subset of SLFN family members (Fig. 5b). We compared the SLFN14 active site arrangement with existing SLFN structures to uncover a very similar organization of the active site core, with the highest degree of overlap observed in the ExxxxExK motif and its adjacent arginine residue (Fig. 5c). In contrast, the SLFN14 aspartate residues, particularly residue D249, do not align well with the equivalent residues in SLFN12, SLFN5, and to a lesser extent

SLFN13 (Fig. 5c). Whether these discrete differences in the active site periphery are sufficient to influence catalytic output or site-specificity remains unknown.

To establish the importance of individual residues within the ExxxxExK motif, we performed an in vitro RNA cleavage study. We generated SLFN14 variants harboring single missense mutations to the consensus ExxxxExK motif. We expressed and purified these SLFN14 variants as described above for wild-type protein and confirmed their purity by SDS-PAGE analysis (Supplementary Fig. 16a). We measured native tRNA[Ser] cleavage using a high enzyme concentration. Glutamate and lysine residues encoded within the ExxxxExK motif were mutated to alanine (E206A, E211A, K213A) and shown to inactivate the SLFN14 ribonuclease (Fig. 5d). Considering all SLFN biochemical studies have relied on alanine mutagenesis, we extended our analysis to include conservative changes to the ExxxxExK motif. Despite preserving amino acid polarity, length, and, in one case, charge, disruptions to the ExxxxExK motif (E206Q, E211Q, K213R) still abolish SLFN14 ribonuclease activity (Fig. 5d). To determine whether SLFN14 enzymatic inactivation was due to a secondary effect, we measured the thermal stability and native tRNA binding activity of SLFN14 ExxxxExK variants (Fig. 5e, Supplementary Fig. 16b). Conservative changes to the ExxxxExK motif did not substantially impact protein stability and only mildly reduced SLFN14 substrate binding affinity. Together, these data underscore the importance of the ExxxxExK motif for RNA cleavage.

Next, we investigated the role of peripheral active site residues. We introduced mutations to SLFN14 R133 that either maintain (R133K) or eliminate (R133A) its positive charge. The R133A mutation inactivates SLFN14, whereas the R133K variant preserves its nuclease activity, suggesting its positive charge plays an important role in coordinating nearby E211 for RNA hydrolysis (Fig. 5d). Interestingly, the SLFN14 R133K variant also exhibits a more stable interaction with native tRNA compared to the wild-type protein (Fig. 5e). Individual point mutations to D248 (D248N, D248A) and D249 (D249N, D249A) moderately impacted native tRNA[Ser] cleavage, confirming earlier work that these SLFN residues play a limited role in tRNA cleavage (Fig. 5d)[16,23,26]. To determine whether these conserved aspartate residues have a substrate-dependent role in nuclease activity, we further analyzed SLFN14 cleavage activity using a larger RNA substrate. The SLFN14 D249N variant retains residual nuclease activity towards native 5S rRNA, whereas the SLFN14 D248A/N and D249A variants show a significantly reduced 5S rRNA cleavage activity (Supplementary Fig. 16c). Collectively, our work demonstrates the conditional, and potential substrate-dependent, roles of peripheral active site residues in SLFN14.

## Structural analogy with prokaryotic restriction endonucleases

The SLFN catalytic center is strikingly similar to ancient restriction endonucleases (REases). Prokaryotic REases, such as BamHI, EcoRI, and EcoRV, are self-defense factors that recognize short palindromic sequences to cleave bacteriophage DNA upon viral infection[48]. While their substrate recognition mechanism is likely distinct from the SLFN family, their catalytic D-(D/E)hK (where h is a hydrophobic residue) motif shares common parallels with the SLFN ExxxxExK motif, which we now define as the E-EhK motif. Both motifs are displayed on uneven strands of a β-sheet where the first and second acidic residues coordinate cations essential for catalysis (Fig. 6)[49]. Moreover, the REases encode a conserved hydrophobic residue followed by an invariant lysine, a molecular feature that is preserved within the SLFN motif of active family members (Fig. 5b). DNA-bound REase structures reveal a cluster of catalytic residues around the metal binding site and scissile phosphate[22,47]. This structural organization aligns with our SLFN14•RNA structure and existing tRNA-bound SLFN11 structures (Fig. 6a)[28]. Many REases also adopt homodimeric assemblies where each juxtaposed D-(D/E)hK catalytic center concertedly cleaves DNA to

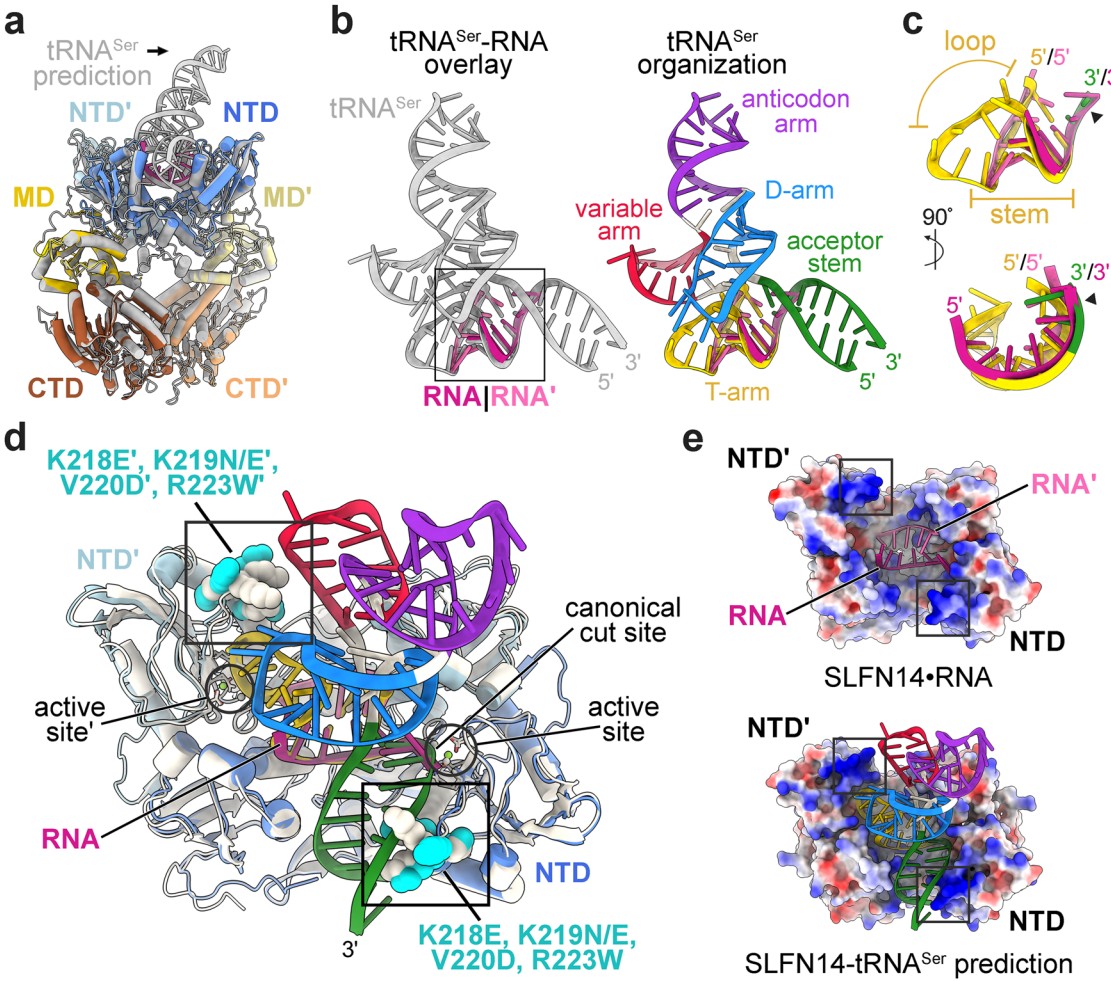

**Fig. 4 | SLFN14-tRNA^Ser prediction model retains the core architecture of the SLFN14•RNA structure. a** Cartoon representation of state 1 SLFN14•RNA cryoEM structure (colored) and SLFN14-tRNA^Ser prediction model generated by AlphaFold3 (gray)[47]. **b** Overlay of the predicted tRNA^Ser and experimental RNA modeled into the RNA cleft (left). The black box marks the location of the 3′-overhang RNA molecule from the cryoEM structure. Topological organization of tRNA^Ser (right) where the acceptor stem (green), D-arm (blue), anticodon arm (purple), variable arm (red), and T-arm (yellow) are shown in color. **c** Superimposition of the RNA from the cryoEM structure (pink) and 3′ features of tRNA^Ser from the prediction model, as shown in panel b. The black arrowhead demarcates the putative tRNA^Ser scissile phosphate. **d** Overlay of SLFN14•RNA (gray) and the SLFN14-tRNA^Ser prediction model (colored), where previously reported SLFN14-related thrombocytopenia mutations are shown as spheres and boxed. **e** Electrostatic surface potential showing the RNA cleft formed by SLFN14 NTD dimerization. The SLFN14-associated disease hotspot is boxed for reference. Surface colors from blue to red represent −20 kT/e to + 20 kT/e, respectively.

produce blunt or staggered products[50]. While this raises the question of whether the associated protomers of dimeric SLFN family members can also facilitate simultaneous tRNA cleavage, current evidence suggests this is unlikely to be the case. The symmetric SLFN11 homodimer adopts an asymmetric conformation upon type-I and type-II tRNA binding, where one protomer is active for RNA cleavage, while the adjacent protomer plays a non-catalytic auxiliary role for tRNA processing[22,28].

## Discussion

Here, we present a cryoEM-derived atomic model of the SLFN14 higher-order assembly, which exhibits similarities and distinctions with other SLFN family members. The crescent-shaped protomer of SLFN14 shares a high degree of homology with existing structures of SLFN family members, underlying a universal core architecture[21–24,28]. While SLFN5 and SLFN13 appear monomeric in solution[23,24], other SLFN protomers oligomerize. SLFN11 harbors N- and C-terminal intermolecular interfaces adopting a dimeric state that is very similar to the SLFN14 head-to-head, tail-to-tail configuration reported in this work[22]. The SLFN12 NTD also dimerizes in a similar configuration; however, its

PDE3A binding partner substitutes for an absent SLFN CTD, enabling stable assembly of a heterotetramer[21]. Although the structural organization of the SLFN14 dimer resembles the SLFN11 homodimer and, to a lesser extent, SLFN12-PDE3A, their mechanisms of nuclease regulation differ. SLFN11 adopts a salt-sensitive monomer-dimer equilibrium that correlates with the distribution between the enzyme's basal and stimulated nuclease state, respectively[22]. In addition, an SLFN11-specific C-terminal phosphorylation site inhibits dimerization, suppressing tRNA cleavage and serving as a potential tunable mechanism for catalytic output[27,28]. SLFN12 on its own exhibits weak ribonuclease activity, coupling SLFN12-PDE3A complex formation with tRNA nuclease activation[21,51]. Conversely, the SLFN14 dimerization interface is hydrophobic, acting as a strong molecular glue for a constitutive homodimer and presumably priming the enzyme for RNA cleavage. During the final stages of preparing this manuscript, a study reports a structure of apo SLFN14, independently confirming its stable dimeric arrangement[52]. It is unclear whether the ribonuclease activity of the stable SLFN14 dimer is further modulated by reversible molecular signals, such as protein phosphorylation, as seen with SLFN11 and SLFN12, and what role

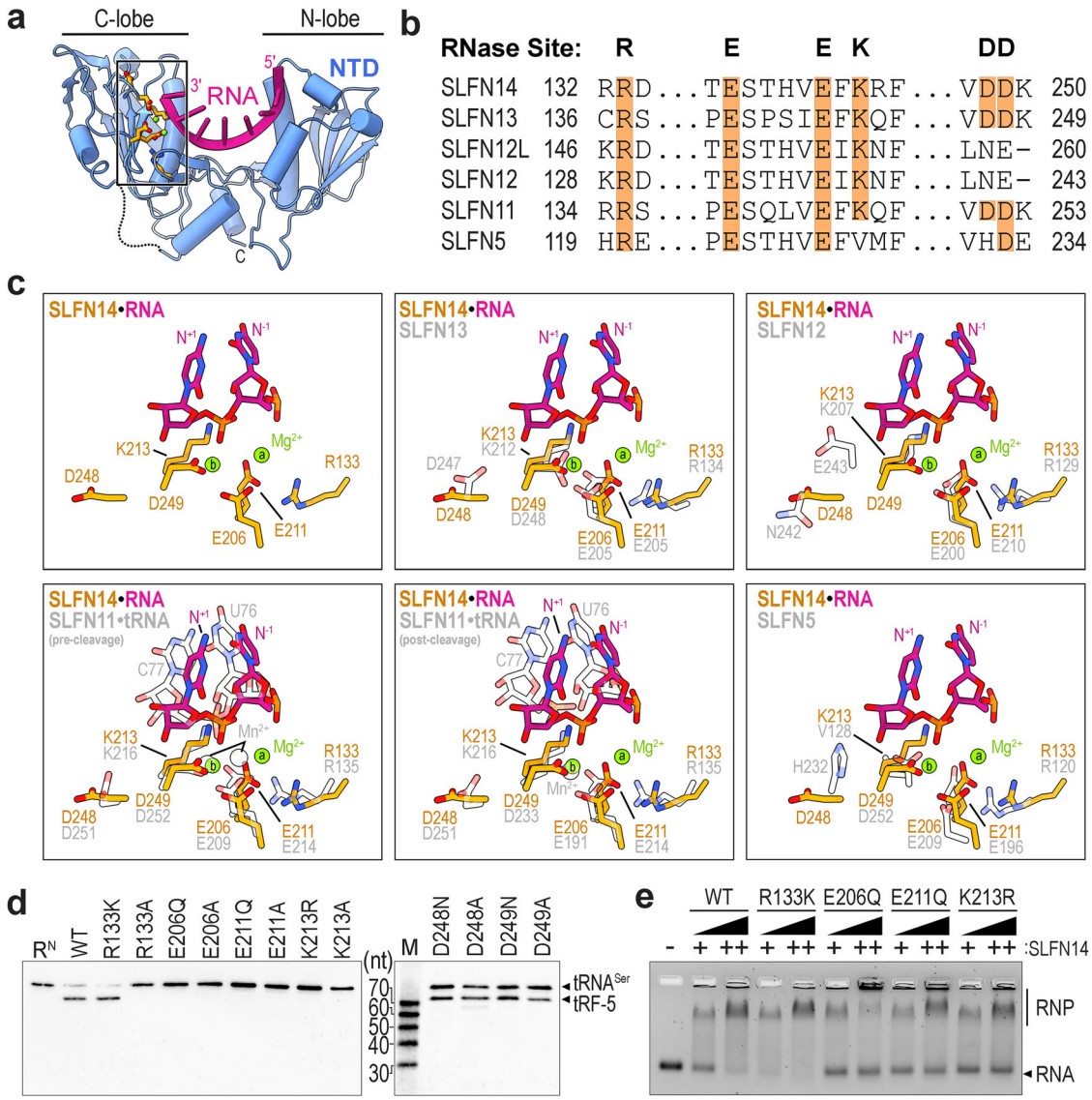

**Fig. 5 | Conserved SLFN14 ExxxxExK motif is critical for RNA cleavage.**
**a** Cartoon representation of a single SLFN14•RNA NTD protomer (blue) where residues clustering around the ExxxxExK motif (orange) are illustrated along with Mg$^{2+}$ metal ions (green) and RNA (pink). **b** Multiple amino acid sequence alignment of human SLFN family members with conserved residues associated with putative ribonuclease function highlighted in orange. Alignments were performed with PROMALS3D[84]. **c** The SLFN metal ion binding site and ExxxxExK motif with surrounding residues. Human SLFN14 (orange) bound to RNA (pink) is superimposed with rat SLFN13 (PDB 5YDO[23]), human SLFN12 (PDB 7LRD[21]), pre-cleavage state human SLFN11-tRNA$^{Leu}$ (PDB 9GMW[28]), post-cleavage state human SLFN11-tRNA$^{Leu}$ (PDB 9GMX[28]), and human SLFN5 (PDB 7Q3Z[24]) shown in gray. **d** Representative

Northern blot of native tRNA$^{Ser}$ (R$^N$, 100 nM total small RNA) processing by SLFN14 variants (400 nM). Black arrowheads mark unprocessed tRNA$^{Ser}$ and cleaved 5′ tRNA$^{Ser}$ fragments (tRF-5). M defines the 70–10-nt RNA ladder with its corresponding nucleotide size on the left. The samples were derived from the same experiment, and the blots were processed in parallel. Three experiments were repeated independently with similar results. **e** Representative electrophoretic mobility shift assay of select SLFN14 variants with native tRNA (0.2 μM total small RNA). The + symbol marks reactions with 2.5 μM protein and ++ denotes mixtures with 5 μM protein. RNP defines SLFN14 ribonucleoprotein complexes and RNA marks free RNA substrate. Three experiments were repeated independently three times with similar results. Source data are provided as a Source Data file.

the SLFN14 homotetrameric species reported in this study may serve in RNA processing[27,28,51]. These remain important questions for future investigation.

Our data reveals the requirement of native tRNA for SLFN14's site-specificity. Our biochemical work demonstrates selective SLFN14 processing at the acceptor stem of many modified native tRNAs, a characteristic that was rarely observed with unmodified synthetic tRNA. SLFN14's striking dependence on modified native RNA for acceptor stem processing was not detected with SLFN11, suggesting it is not a universal requirement for all SLFN family members. We propose that modification(s) encoded within native tRNA is likely directing site-specificity of the SLFN14 endoribonuclease either through a direct or indirect mechanism. The antiviral SAMD9 tRNA

endoribonuclease exerts substrate specificity through recognition of the 2′-*O*-methylation at the wobble position of phenylalanine tRNA (tRNA$^{Phe}$), underscoring the precedent for a direct mechanism where a discrete modification licenses the nuclease for site-specific tRNA cleavage[53]. Intriguingly, we identified a structural analogy between the SLFN catalytic C-lobe and the SAMD9 effector domain[41]. Like the C-lobe, the SAMD9 effector domain also harbors metal-dependent endoribonuclease activity associated with a cryptic E-EhK motif[53,54]. Moving forward, it will be important to discern whether SLFN14 and its family members directly probe for the presence of universally conserved or codon-specific tRNA modifications for substrate recognition or site-specificity. Angiogenin is another stress-activated tRNA nuclease that is regulated by RNA modification[55–57]. tRNA modifications,

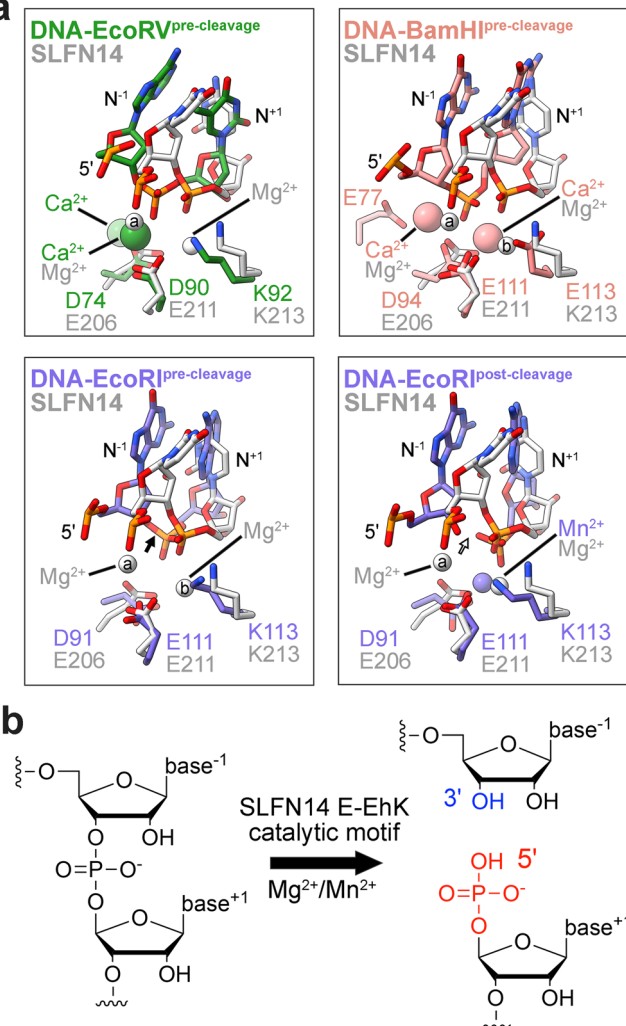

**Fig. 6 | SLFN14 active site shares structural analogy with type-II restrictions factors. a** Overlay of the SLFN14•RNA catalytic center (SLFN14, gray) with pre-transition state type-II restriction factors EcoRV (green, PDB 2BOD[85]), BamHI (pink, PDB 2BAM[86]), and EcoRI (purple, PDB 1CL8[87]), as well as post-transition state EcoRI (purple, PDB 1QPS[87]). Black arrow marks the 3' bridging O-P bond and the white arrow identifies the cut site. **b** Cartoon schematic of RNA cleavage by the SLFN14 ribonuclease using ChemDraw 23.1.2. The SLFN14 E-EhK motif, comprised of E206, E211, and K213, catalyzes the $Mg^{2+}/Mn^{2+}$-dependent cleavage reaction that results in 5'-phosphate (red) and 3'-hydroxyl (blue) RNA ends.

nonspecific RNA processing. The periphery of the C-lobe is decorated with basic residues that contribute to the SLFN14 disease hotspot and correspond to similar RNA-binding residues observed in existing SLFN11 structures[28]. This positively charged surface at the RNA cleft entrance may facilitate SLFN14 substrate recruitment and could dysregulate target selectivity in SLFN14-related thrombocytopenia patients. Upon entering the RNA cleft, the phosphodiester backbone of duplexed RNA is buttressed by the juxtaposed N-lobes, running its helical axis across the elongated cleft base. A prediction model comprising intact tRNA aligns the phosphodiester backbone of the T-stem with our cryoEM-derived model RNA at the N-lobe RNA binding interfaces and overlays our modeled 3'-overhang with the acceptor stem at the C-lobe active site. Likewise, recent tRNA-bound structures of SLFN11 reveal electrostatic interactions between its N-lobes and the phosphodiester backbone of the T-arm and acceptor stem of type-I and type-II tRNAs[28]. In the case of tRNA processing, we suspect that the SLFN14 dimer adopts an asymmetric conformation upon tRNA capture, as seen with SLFN11[28], where each protomer recognizes unique features of the T-arm, acceptor stem, and potentially the variable arm. The subtle wobbling of the SLFN14 C-lobes likely provides conformational malleability to the RNA cleft and may be important for accommodating different RNA substrates or aligning the scissile phosphate to the catalytic center. SLFN14 relies on its C-lobe encoded E-EhK catalytic motif, where one of the two protomers likely adopts an activated state for metal-dependent cleavage of the acceptor stem, an event that is likely further regulated by tRNA modifications (Fig. 6b). Collectively, this study provides a framework for understanding the molecular basis underpinning the SLFN14 endoribonuclease in stress-induced tRNA processing.

## Methods

### Molecular cloning of SLFN protein expression vectors
Please refer to Supplementary Data 1 for a list of all expression plasmids used in this study. Full-length human SLFN14 (residues 1–912), along with a downstream tandem Myc-Flag epitope sequence, was inserted into the mammalian expression vector pCMV6-entry between SgfI and PmeI (pMP789; Origene, catalog: RC226257). All SLFN14 variants were created by Q5 site-directed mutagenesis (NEB) using pMP789 as a template. Full-length human SLFN11 (residues 1–901), along with a downstream tandem Myc-Flag epitope sequence, was inserted into the mammalian expression vector pCMV6-entry between SgfI and PmeI (pMP1160; Origene, catalog: RC226247). The SLFN11 E214A variant was synthesized by GenScript (Piscataway, NJ) using pMP1160 as a template. All plasmids were verified by DNA sequencing.

### SLFN14 and SLFN11 Expression and Purification
Plasmids encoding SLFN14 and SLFN11 variants were transfected into HEK293 cells adapted to grow in suspension (Thermo Scientific, catalog: R79007). Cells were transfected with 1 μg/mL of purified plasmid DNA and 2 μg/mL of polyethylenimine (Polysciences Inc., catalog: 24765). Cells were harvested 48–72 h post-transfection and frozen at −80 °C. Transfected cells were resuspended in lysis buffer (50 mM Tris pH 7.4, 600 mM NaCl, 5 mM MgCl₂, 0.5% Triton-X100, 1 mM benzamidine, 1 mM phenylmethylsulfonyl fluoride, 5 μg/mL leupeptin, 0.7 μg/mL pepstatin A, 0.25 μ/μL GENIUS nuclease (Santa Cruz, catalog: sc-391121) and lysed with gentle rocking at 4 °C for 30 min or sonicated at 750 watts, 50% amplitude for 5 min (30 second pulses) at 4 °C (Qsonica). Samples were clarified at 15,871 x g for 20 min at 4 °C. Clarified lysate was incubated with anti-Flag M2 affinity gel (Sigma, catalog: A2220) for 1 h at 4 °C. The lysate mixture was transferred to a micro Bio-spin column (Bio-Rad) and the resin was washed with 1 mL of Wash 1 buffer (50 mM Tris pH 7.4, 600 mM NaCl, 5 mM MgCl₂, 0.1% Tween-20), 1 mL Wash 2 buffer (50 mM Tris pH 7.4, 500 mM NaCl, 5 mM MgCl₂, 0.1% Tween-20), 1 mL Wash 3 buffer (50 mM Tris pH 7.4,

particularly methylation, protect tRNA sites, like the anticodon arm, from Angiogenin processing[58–61]. Since we only observed SLFN14 processing outside of the acceptor stem when using synthetic tRNA, it is possible that tRNA modifications shield non-cognate sites, such as the tRNA^Ser anticodon arm, from spurious processing. In the absence of a molecular signal, Angiogenin is a relatively inefficient endoribonuclease with low specificity towards tRNA[55–57,62]. Recent work uncovers that the ribosome is the activator and target specificity factor for Angiogenin's tRNA cleavage activity[63]. Our in vitro analysis of SLFN14 demonstrates relatively low and nonspecific tRNA cleavage activity compared with its SLFN11 family member, prompting the question of whether SLFN14's previously reported association with the ribosome plays a role in its endoribonuclease stimulation and selectivity[16].

Our data provides molecular insights into SLFN14 substrate binding and processing. SLFN14 dimerization forms a composite RNA cleft that sterically restricts access to the N-lobe RNA binding surfaces and recessed C-lobe ribonuclease catalytic centers, likely limiting

500 mM NaCl, 5 mM $MgCl_2$, 0.1% Tween-20, 10 mM ATP), and 0.25 mL Wash 2 buffer. SLFN14 and SLFN11 variants were eluted with 3x flag peptide (Thermo Scientific, catalog: A36805) by incubating the resin with elution buffer (50 mM Tris pH 7.4, 500 mM NaCl, 5 mM $MgCl_2$, 0.1% Tween-20, 300 ng/mL 3x flag peptide) for 30–60 min at 4 °C. A Zeba spin desalting column (Thermo Scientific, catalog: A57759) was used to remove excess 3x flag peptide and exchanged for storage buffer (20 mM Tris pH 7.4, 150 mM NaCl, 5 mM $MgCl_2$, 1 mM TCEP). Purified SLFN14 and SLFN11 variants were flash-frozen in liquid nitrogen and stored at −80 °C for future use. Purified SLFN14, SLFN14$^{ΔCTD}$, SLFN14$^{NTD}$, and SLFN14$^{CTD}$ variants (5–10 μM) were analyzed using a Superdex-200 Increase 3.2/300 (Cytiva) column equilibrated with storage buffer and calibrated with gel filtration protein standards (Bio-Rad, catalog: 1511901). Peak fractions were resolved by SDS-PAGE and visualized using silver stain or Coomassie G-250 stain, where indicated. Two gel filtration experiments were repeated independently for each construct with similar results. Source data are provided as a Source Data file.

Purified recombinant SLFN11 and SLFN14 protein was confirmed by liquid chromatography tandem mass spectrometry (LC-MS/MS) within $n = 1$ sample/independent experiment. The isolated proteins (SLFN11 and SLFN14) from HEK293 cells were resolved using an SDS-PAGE gel (NuPAGE 10% Bis-Tris Gel, Invitrogen). The protein band of interest at around 100 kDa was excised from the gel and subjected to in-gel digestion using trypsin and LysC protease mix. The extracted peptides were dried in a speed vac and dissolved in 5% methanol containing 0.1% formic acid buffer. The peptide samples were analyzed using a nanoLC1200 system coupled to Orbitrap Fusion Lumos mass spectrometer (Thermo Scientific, San Jose, CA). The MS data was acquired for a 110-min total run time at 200 nl/min using the data-dependent mode. The full MS scan was acquired in Orbitrap in the range of 350–1500 m/z at 120,000 resolution, followed by MS2 in Ion Trap (HCD 32% collision energy, maximum injection time 30 ms). The dynamic exclusion was set to 15 s. The MS raw data was searched in Proteome Discoverer 2.0 software (Thermo Scientific, San Jose, CA) with the Mascot algorithm (v2.5) against the human NCBI refseq protein database (updated 2021_12_23). The precursor ion tolerance and product ion tolerance were set to 20 ppm and 0.5 Da respectively, and a maximum of two missed cleavages with trypsin was allowed. The following variable modifications were allowed: Oxidation on methionine, protein N-terminal acetylation, Carbamidomethylation on cysteine, GlyGly remanent on lysine, Acetylation on lysine, Methylation on arginine and Phosphorylation on serine, threonine and tyrosine. The peptides identified from mascot result file were validated in Percolator with 5% false discover rate (FDR). The peptide spectra were manually validated for correct site assignment.

### Preparation of Synthetic and Native RNA Substrates

An RNA ladder was prepared by mixing equimolar 5′-Cy5 labeled commercial RNA oligonucleotides (Dharmacon) ranging from 70-nt to 10-nt in length with 10-nt intervals. The resulting RNA ladder is herein the 70–10-nt RNA ladder. See Supplementary Data 2 for RNA sequences.

Unlabeled synthetic RNA substrates were synthesized using recombinant T7 polymerase (NEB HiScribe, catalog: E2040) and blunt-ended double-stranded DNA encoding the T7 promoter immediately followed by the desired tRNA gene (Integrated DNA Technologies). See Supplementary Data 3 for DNA template sequences. The resulting enzymatic reactions were treated with DNase I, and synthetic RNA was purified using the RNA Clean & Concentrator kit (Zymo Research, catalog: R1014). Purified synthetic RNA was quantified using a Qubit 4 fluorometer. Synthetic RNA was resolved on commercial 15% polyacrylamide (8 M urea) gels (Bio-Rad, catalog: 4566055) to verify a single RNA band and confirm the expected size using the 70-10nt RNA ladder.

Cellular RNAs (herein native RNA) were enriched from HEK293 cells (Thermo Scientific, catalog: R79007). Long native RNAs were isolated using Trizol reagent (Invitrogen, catalog: 15596026), treated with DNase I, and purified using the RNA Clean and Concentrator kit (Zymo Research, catalog: R1017) to enrich for human 28S and 18S rRNA. Short native RNAs (>200-nt) were extracted using the PureLink miRNA Isolation Kit (Invitrogen, catalog: K157001), treated with DNase I, and purified using the RNA Clean and Concentrator kit (Zymo Research, catalog: R1014) to enrich for 5S rRNA and tRNAs. Native RNAs were quantified at an absorbance of 260 nm. Enriched long and short native RNAs were resolved by 1% formaldehyde-agarose gels and commercial 15% polyacrylamide (8 M urea) gels (Bio-Rad, catalog: 4566055), respectively, to verify the absence of RNA degradation and confirm the expected size using nucleic acid ladders.

### In vitro RNA binding assay

SLFN14 RNA binding activity was measured by electrophoretic mobility shift assays (EMSA). Native short RNA (0.2 μM) was incubated in the absence and presence of SLFN14 variants (2.5 μM and 5 μM) for 1 h at 4 °C in the presence of 10% glycerol. Samples were resolved on a 1% agarose gel with 1x Tris-acetate-EDTA running buffer and visualized on a ChemiDoc MP system (Bio-Rad) using 1x GelRed stain (Millipore Sigma, catalog: SCT123). Representative gels are shown from three independent replicates. An uncropped and unprocessed image of the representative gel is shown in the Source Data file.

### In vitro RNA cleavage assays

For in vitro cleavage of native 28S and 18S rRNA, long native RNA (1 μg) was mixed with purified SLFN14 variants (0.25, 0.5, 1 or 2 μM) in RNA cleavage buffer (20 mM Tris pH 7.4, 100 mM NaCl, 1 mM $MgCl_2$, 1 mM TCEP). Mixtures were supplemented with 50 mM EDTA and 1 mM $MgCl_2$ or $MnCl_2$, where indicated. Protein mixtures were incubated at 37 °C for 30 min. Reactions were stopped with an equal volume of formamide-loading dye. Mixtures were resolved by 1% formaldehyde-agarose gels (supplemented with 0.25 μg/mL ethidium bromide) with 1x MOPS running buffer and visualized on a ChemiDoc MP system (Bio-Rad) using the ethidium bromide setting.

For in vitro cleavage of 5S rRNA and tRNAs, synthetic RNA (100 nM) or total short native RNA (100 nM) were incubated with SLFN14 and SLFN11 variants (50–400 nM) for 40 min at 37 °C. Where specified, mixtures were supplemented with $16 − 2 × 10^{−4}$ units of yeast XRN-1 exoribonuclease (NEB, catalog: M0338). The reactions were stopped with urea-loading dye and incubated with 25 mM EDTA, 1 mg/mL proteinase K (NEB, catalog: P8107) for 15 min at room temperature. Mixtures were resolved by commercial 15% polyacrylamide (8 M urea) gels (Bio-Rad, catalog: 4566055) with 1x Tris-borate-EDTA running buffer. Gels were either stained with SYBR Gold (Thermo Scientific, catalog: S11494) or analyzed by Northern blot analysis[64]. Resolved RNA cleavage reactions were transferred to Hybond-N + membrane (Cytiva, catalog: RPN203B) using a semi-dry Trans-Blot Turbo Transfer System (Bio-Rad). Membranes were chemically crosslinked using 0.16 M 1-Ethyl-3-(3-dimethylaminopropyl) carbodiimide (EDC) at 60 °C for 1 h and blocked with ULTRAhyb ultrasensitive hybridization buffer (Invitrogen, catalog: AM8669) at 37 °C for 30 min. Membranes were incubated with 1 nM biotinylated DNA probe overnight at 37 °C. See Supplementary Data 3 for DNA probe sequences. Following the manufacturer's instructions, Northern blots were imaged using the Chemiluminescent Nucleic Acid Detection Module Kit (Thermo Scientific, catalog: 89880). Uncropped and unprocessed blots of the representative RNA cleavage assays are shown in the Source Data file.

### Cryo-EM specimen preparation, data collection, and image processing

Purified SLFN14 was applied to a Zeba spin desalting column (Thermo Scientific, catalog: A57759) to exchange for EM buffer (20 mM Tris pH

**Table 1 | Cryo-EM data collection, refinement, and validation statistics**

|  | SLFN14 | SLFN14 + tRNA | SLFN14 + tRNA |
|---|---|---|---|
| *Data collection and processing* | | | |
| Magnification (×) | 105,000 | 105,000 | 105,000 |
| Voltage (kV) | 300 | 300 | 300 |
| Electron exposure (e⁻/Å²) | 50.2 | 50.2 | 50.2 |
| Defocus range (μm) | −1.0 to −2.2 | −1.0 to −2.2 | −1.0 to −2.2 |
| Pixel size (Å) | 0.416 | 0.416 | 0.416 |
| Stage tilt (°) | 0 | 0 | 25 |
| Micrographs | 5024 | 4351 | 1692 |
| ***Reconstruction*** | | **RNA•SLFN14 State 1 (PDB 9NYY) (EMD-49946)** | **RNA•SLFN14 State 2 (EMD-49947)** |
| Symmetry imposed | | C1 | C1 |
| Initial particle images (no.) | | 10,302,578 | 10,302,578 |
| Final particle images (no.) | | 338,469 | 188,010 |
| Map resolution (Å) (FSC = 0.143) | | 2.73 | 3.11 |
| Map resolution range (Å) | | 2.34–6.56 | 1.78–7.86 |
| *Refinement* | | | |
| *Model composition* | | | |
| Chains | | 4 | |
| Non-hydrogen atoms | | 13,446 | |
| Protein residues | | 1643 | |
| Nucleotides | | 12 | |
| Ligands | | 3 ($Mg^{2+}$), 2 ($Zn^{2+}$) | |
| *B factors (Å²)* | | | |
| Protein | | 47.41 | |
| Nucleotide | | 31.80 | |
| Ligand | | 51.24 | |
| *Map-model CC* | | | |
| CC (mask) | | 0.76 | |
| CC (volume) | | 0.77 | |
| CC (peaks) | | 0.75 | |
| CC (box) CC (ligands) | | 0.78 0.77 | |
| *R.M.S. deviations* | | | |
| Bond length (Å) | | 0.003 (0) | |
| Bond angles (°) | | 0.580 (0) | |
| *Validation* | | | |
| MolProbity score | | 2.10 | |
| Clashscore | | 7.79 | |
| Poor rotamers (%) | | 2.83 | |
| *Ramachandran plot* | | | |
| Favored (%) | | 95.29 | |
| Allowed (%) | | 4.71 | |
| Disallowed (%) | | 0.00 | |

7.4, 150 mM NaCl, 1 mM $MgCl_2$, 1 mM TCEP). SLFN14 sample (0.6 mg/mL) was incubated with and without purified tRNA from brewer's yeast (Thermo Scientific, catalog: AM7119) at a 1:1 molar ratio and applied to a glow-discharged (15 mA current for 20 s) 200-mesh QUANTIFOIL R1.2/1.3 2 nm ultra-thin carbon grids (Electron Microscopy Sciences, catalog: Q250CR1.3–2 nm). Vitrification was performed using the Vitrobot (Thermo Scientific) with a 0.5-second blot time and 100% humidity. Images were collected on a Titan Krios operated at 300 keV and equipped with a K3 Summit direct electron detector (Gatan) and Quantum energy filter (Gatan) using Serial EM version 3.8.16 and EPU

3 software. Movie stacks were collected at a nominal magnification of 105,000 × in super-resolution mode with a pixel size of 0.416 Å. A total dose of 50.2 e⁻/Å² was fractionated across 40 frames under super-resolution mode. Defocus values ranged from − 1.0 to − 2.2 μm. Overall, 5,024 movies were collected in the absence of tRNA, 4,351 movies were collected in the presence of tRNA, and 1,692 movies were recorded with tRNA and a 25° stage tilt. The datasets were merged for cryoEM reconstructions. Data collection statistics are listed in Table 1.

Beam-induced motion was corrected using MotionCor2[65] with 2-fold binning, and CTF parameters were calculated using CTFFIND4[66] from aligned dose-weighted images. A total of 10,302,578 particles were automatically picked using RELION 4.0[67–69] from 11,067 images and imported into cryoSPARC v4.2.1 and v4.6.2[70]. From 300 two-dimensional classes, 85 classes containing 1,120,926 particles were selected for ab initio three-dimensional reconstruction, which produced one good class with recognizable structural features and two bad classes that did not have structural features. Both the good and bad classes were used as references in heterogeneous refinement (cryoSPARC v4.2.1), which yielded a good class (957,812 particles) at 3.9 Å resolution. Non-uniform refinement and CTF refinement (cryoSPARC v4.2.1) were performed with an adaptive solvent mask. Further 3D classification was performed in RELION 4.0 and found two classes in different, albeit subtle, conformations. After particle polishing, non-uniform refinement with C1 symmetry, and CTF refinement, two asymmetric maps with an overall resolution of 2.73 Å (state 1) and 3.11 Å (state 2) were yielded with 338,469 and 188,010 particles, respectively. Resolutions were estimated using the gold-standard Fourier shell correlation with a 0.143 cut-off[71] and high-resolution noise substitution[72]. Local resolution was estimated using ResMap[73] (cryoSPARC v4.2.1), and map sharpening was performed using DeepEMhancer 0.13[74].

## Model building and refinement

Overlay of state 1 and state 2 maps reveals subtle conformational changes at the N- and C-termini. The catalytic C-lobes are poorly defined in the state 2 map, limiting accurate model building. The state 1 cryo-EM reconstruction has an estimated resolution of 2.73 Å. At this resolution, we can observe clear density for the SLFN14 NTD, MD, and CTD across both protein chains, although protomer B is less resolved than protomer A. We also observe side chain density, density within the RNA cleft, and poor density for flexible loops. For the state 1 reconstruction, we used the human SLFN14 AlphaFold model prediction as an initial model[75]. The SLFN14 initial model was processed and docked into the state 1 cryoEM reconstruction and rebuilt using Phenix 1.21-1-5286[76]. Manual adjustments were made to the model in COOT 0.9.8.94, followed by real-space refinement[76,77]. Flexible loops with ambiguous density were omitted. Density within the dimeric SLFN14 RNA cleft was modeled as RNA; however, the density lacks nucleobase features. For this reason, the 3′-overhang duplexed RNA molecule was modeled with an arbitrary sequence; chain C: 5′-AUGGG-3′ and chain D: 5′-CCCACUC-3′. Molprobity was used to evaluate the model, and statistics are listed in Table 1[78]. The reconstruction of state 1 was more clearly resolved for the active site in protomer A (chain A) compared to protomer B (chain B). For this reason, structural analysis and figure generation of the SLFN14 nuclease active site were performed using protomer A. AlphaFold3 generated a tRNA^Ser-bound prediction model for SLFN14 using human serine tRNA (anticodon TGA) 2-1 obtained from RNAcentral and human SLFN14 protein sequence from the UniProt database[47,79,80]. Maps and models were visualized using UCSF Chimera 1.17.3, UCSF ChimeraX 1.8, and Pymol v3.1.3[81,82].

## Reporting summary

Further information on research design is available in the Nature Portfolio Reporting Summary linked to this article.

## Data availability

The data supporting the findings of this study are available from the corresponding authors upon request. Source data for the figures and Supplementary Figures are provided as a Source Data file. The cryoEM maps and the atomic coordinates generated in this study have been deposited in the EMDB and PDB under accession codes EMD-49946, EMD-49947, and PDB 9NYY. Previously published accession codes are deposited in the PDB as follows PDB 5YD0, 7LRD, 9GMW, 9GMX, 7Q3Z, 2B0D, 2BAM, 1CL8, 1QPS, 7ZEL, and 9ERF. The mass spectrometry proteomics data have been deposited to the ProteomeXchange Consortium via the PRIDE partner repository with the dataset identifier PXD064237. Source data are provided in this paper.

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

## Acknowledgements

We thank Dr. Robin Stanley and Dr. Kathleen McCann for their critical reading of this manuscript. We are grateful to Dr. Calla Olson, Dr. Thomas Westbrook, and Dr. Jeremy Wilusz for providing reagents and access to instrumentation. This work was supported by the US National Institute of Health Extramural Research Program, National Institute of Environmental Health Sciences (NIEHS; R00ES030735 to M.C.P.), and National Institute of General Medical Sciences (NIGMS; R35GM147123 to M.C.P.). This study was also supported by the Cancer Prevention and Research Institute of Texas (CPRIT RR200076 to M.C.P) J.V.R. was supported by a training fellowship from the Gulf Coast Consortia, on the Houston Area Molecular Biophysics Program (Grant No. T32 GM008280 to J.V.R.). CryoEM data was collected at the Baylor College of Medicine CryoEM ATC, subsidized by the CPRIT Core Facility Award RP190602, which also supported the acquisition of CryoEM equipment used in this study. We are grateful to all the members of the Baylor College of Medicine cryoEM core for their support in single-particle EM screening and collection. We thank Dr. Gaya P. Yadav for cryoEM data collection at the Laboratory for Biomolecular Structure and Dynamics (LBSD) of Texas A&M University. The LBSD is supported, in part, by the Department of Biochemistry & Biophysics, AgriLife, and Texas A&M University. We are grateful to the Baylor College of Medicine Mass Spectrometry Proteomics Core for their assistance in protein identification by mass spectrometry. BCM Mass Spectrometry Proteomics Core is supported by the Dan L. Duncan Comprehensive Cancer Center NIH award (P30 CA125123), CPRIT Core Facility Award (RP210227).

## Author contributions

J.V.R., A.M.C., H.E.S., S.M. and M.C.P. conceived, designed, and executed all studies. J.V.R., L.W. and M.C.P. performed EM image processing. J.V.R. and M.C.P. performed model building and refinement. M.C.P. wrote the manuscript, which was edited and approved by all authors.

## Competing interests

The authors declare no competing interests.
