## [Transparent Peer Review file · Nature Communications]

CryoEM structure of the SLFN14 endoribonuclease reveals insight into RNA binding and cleavage

Corresponding Author: Dr Monica Pillon

Version 0:

Reviewer comments:

Reviewer #1

(Remarks to the Author)

Van Riper et al. examined the cryo-EM structure and endonucleolytic activity of human endoribonuclease SLFN14 in their manuscript.

SLFN14, a post-transcriptional regulator, plays a role in RNA degradation, particularly ribosomal RNA (rRNA), transfer RNA (tRNA), and messenger RNA (mRNA). Its nuclease activity has been associated with antiviral defense mechanisms and platelet function. Despite this, the molecular mechanisms regulating SLFN14 substrate recognition and catalytic activation have remained unclear. Dysregulation of SLFN14 is implicated in human diseases, including ribosomopathies and inherited thrombocytopenia, underscoring the need to understand its RNA processing activity and functional mechanism.

Recently the Schlafen family of proteins has got attention among researchers seeking to understand the detailed mechanism of this conserved protein family. While X-ray and cryo-EM structures of human SLFN12, SLFN11, SLFN5, and partially SLFN13 have been solved, providing reasonably accurate AlphaFold predictions for all family members in monomeric and dimeric forms, human SLFN14 has been the least studied protein to date. Its biochemical and structural characterization represents an important advancement for the field.

Unfortunately, the resolution of the cryo-EM structure is relatively low, preventing detailed molecular conclusions. Conserved active site residues for all SLFN proteins have been proposed in earlier publications. From Supplementary Figure 4, it is unclear whether the authors attempted masked refinement of the N- and C-terminal domains separately to enhance the resolution of individual domains. Could additional data collection enhance the resolution of the structure?

While it is assumed that the active site of SLFN14 largely resembles that of other SLFNs e.g. SLFN11 and 12, the paper lacks a direct structural comparison and overlay of the active sites. This comparison could shed light on the similarities and differences in substrate recognition, especially between SLFN11 and SLFN14. The structure of SLFN11 complexed with tRNA has been solved. Did the authors attempt to solve complex structures with the identified SLFN14 substrates, or is that feasible within the revision timeframe?

SLFN14's ribonuclease activity depends on magnesium ions in contrast to SLFN11, which is solely active with manganese. Could the authors test SLFN14's ability to cleave substrates with other metal ions, such as manganese, and determine if different ions lead to distinct cleavage products? The high protein concentration used in all cleavage experiments, as evident in e.g. Figure 3, suggests SLFN14 may be a relatively unproductive enzyme. Could the authors please explain this behavior?

While the nuclease assays in Figures 1 and 3 are convincing and include necessary controls, the fragmented tRNA experiment in Figure 2c-d is puzzling. The tRNA cartoon is a simplified representation, and dividing tRNA into minimal pieces lead to a different folding of the individual parts. Hence this experiment may lead to incorrect assumptions. Therefore, it is advisable to remove the entire experiment. Could the authors consider sequencing the tRNA cleavage products instead?

In Supplementary Figure 2, the authors demonstrate SLFN14's ability to bind single-stranded DNA (ssDNA). In SLFN11, ssDNA binds on top of the C-terminal helicase domain. Could the authors use mutational or structural studies to pinpoint where SLFN14 binds ssDNA?

The presented data lead to a new tRNA binding model in Figure 7, differing from the SLFN11 tRNA binding mode. A detailed comparison of the residues involved in substrate recognition in both proteins would provide insights into the differences among the structurally conserved but functionally diverse Schlafen family members.

In summary, the findings are novel and significant in the field of the Schlafen family of proteins. While the work provides valuable insights, enhancing the structure's quality or presenting complex structures of SLFN14 with tRNA substrates would strengthen the validity of the proposed model.

Minor comments:

In Figure 3e, indicating the labels 3'fluorescein and 5'Cy5 in the tRNA cartoon would help the reader. It was not possible to play Video 3.

Reviewer #2

(Remarks to the Author)

In their manuscript, Van Riper et al. perform a functional and structural characterization of human SLFN14, a member of the Schlafen (SLFN) family of endoribonucleases. Using recombinantly expressed and purified full-length protein, the authors report a cryo-EM reconstruction of the SLFN14 at 5.5 Å resolution, which reveals a homodimeric architecture and conformational flexibility. Moreover, the authors perform an in vitro biochemical characterization of SLFN14, demonstrating a magnesium-dependent ribonuclease activity, depending on an ExxxxExK catalytic motif, consistent with previous analyses of other SLFN homologs. Notably, these analyses reveal the ability of SLFN14 to cleave tRNAs in the anticodon loop rather than the acceptor stem, which leads the authors to propose that SLFN14, in notable contrast to other previously characterized SLFN homologs, is an anticodon-stem-loop-specific endoribonuclease.

Overall, this work is a timely and potentially important contribution to the field of tRNA-cleaving ribonucleases and their diverse roles in gene-regulatory circuits. In particular, the proposed mechanism of anticodon-specific cleavage by SLFN14, if true, would be a critical novel insight. The experimental work is well-executed. A minor caveat is the relatively low resolution of 5.5 Å of the reported cryo-EM reconstruction, which, however, appears to be of sufficiently high quality to support the main claims regarding the structure made by the authors.

One major concern is insufficient reference to earlier work, both regarding the mechanistic basis for SLFN-mediated RNA cleavage in general, but in particular earlier structural work. Metzner et al. previously reported a 2.86 Å reconstruction for the human SLFN11 homodimer (PMCID: PMC9482658), which appears to be highly similar to the one reported here for human SLFN14. Although a direct structural comparison between the two SLFN homologs (and the structure of other homologs) would be highly relevant, such a comparison is missing completely. Given that the authors propose a radically new mode of tRNA binding and cleavage, such a comparison would be informative as to how these two closely related enzymes achieve such distinct substrate recognition and cleavage mechanisms. Also, the authors assert some observations in a manner that they appear novel. For example, in lines 310-313 the authors write: "Multiple sequence alignments across the SLFN family identify a consensus ExxxxExK (where x is any residue) motif harboring two invariant glutamates and a well-conserved lysine residue (Fig. 1b). We define this discrete element as a putative catalytic motif...". This is not an original observation, but has been noted previously (e.g., PMCID: PMC5862951, PMC9256597, PMC9482658). The authors should take care to mention the previous work where appropriate.

The second major concern regards the claim that SLFN14, in stark contrast to all other previously characterized SLFN homologs, is anticodon-stem-loop specific: "Our work shows that SLFN14 is not an acceptor stem nuclease but an anticodon stem-loop nuclease" (line 617). Thus, the authors specifically exclude the possibility that SLFN14 cleaves the acceptor-stem at all. This claim is based exclusively on in vitro cleavage assays with synthetic, unmodified RNAs, labeled on one or both ends with bulky fluorophores. Below, I elaborate on several specific points.

1. The analysis starts with cleavage assays on small RNAs isolated from human cell lines (Fig. 1f and Suppl. Fig 1b). These indeed show formation of short, ca. 30 nt long fragments; however, the bands with the highest intensity correspond to ca. 65 nt. This would agree with a cleavage position at the base of the acceptor stem of type I tRNAs (corresponding fragments from type II tRNAs would run together with full-length type I tRNAs). The Northern blot results in Suppl. Fig. 1d appear to further confirm this, showing strong bands immediately below the full-length tRNAs. No bands are visible for tRNA-halves. The authors should explain a) how the longer fragments are generated if not by acceptor-stem cleavage, and b) why there are no ~30 nt bands in the northern blots in Suppl. Fig. 1. c) If this is due to the characteristics of the probes used for northern blotting, the authors should repeat the experiment at least for some of the tRNAs with probes complementary to the 30 nt 5'-fragments. This will be important to demonstrate that SLFN14 not only cleaves the anticodon on unmodified synthetic tRNAs, but also on mature and fully modified tRNAs. d) Is there a rationale as to why the authors focus exclusively on the small fragments (~30 nt) and completely disregard the longer (~65 nt) fragments in their biochemical characterization?

2. The rationale to use mitochondrial tRNAMet as model substrate remains unclear. Given that the authors claim to demonstrate "an essential role for the tRNA elbow, a universal structural feature formed through tertiary interactions between the D-loop and T-loop" (line 381), it is important to note that mt-tRNAMet does not contain a canonical elbow and may not form any of those D-loop-T-loop tertiary interactions that define the classical cytosolic tRNA elbow. Moreover, unless SLFN14 enters into the mitochondria, its cleavage pattern of mt-tRNAMet is unlikely to be physiologically relevant at all. Thus, unless there is a specific reason to use it, mt-tRNAMet is a poor choice as a model substrate for SLFN14 (the reference to its use by Metzner et al (PMCID: PMC9482658) appears unmotivated). The authors should use a more likely and thus physiological relevant substrate, such as a eukaryotic cytoplasmic type I tRNA (see below).

3. To demonstrate dependency on the tRNA elbow, the authors perform cleavage assays with 5'Cy5-labeled mt-tRNAMet fragments of different length (Fig. 2c-d). They conclude that cleavage depends on the elbow (line 381), albeit indirectly (line 640). However, their binding model (Fig. 7a) still begs the question of how this indirect effect comes about and why the anticodon-stem-loop substrate is not cleaved at all (Fig. 2c-d). Moreover, they conclude that the acceptor stem is

dispensable (line 377). This, of course, is only true for anticodon-cleavage, which they focus on. It does not demonstrate that SLFN14 does not interact with the acceptor stem in a more SLFN11-like manner and may even cleave it on unlabeled or/and modified substrates. To demonstrate this, the authors should do cleavage and binding assays with the same tRNA fragments (or preferably with those from a cognate substrate instead of the mt-tRNA) in the absence of fluorophores (or repeat the same pattern with 3'-label).

4. A central problem is that the claim of anticodon-specificity is based entirely on in vitro cleavage assays using chemically synthesized tRNAs with one or two bulky fluorophores on their 5' and 3' ends. a) These fluorophores may interfere with productive acceptor stem binding by SLFN14, thereby promoting a (possibly otherwise less-favored) cleavage in the anticodon. b) Position 32 in the anticodon-loop is modified in the majority of cytosolic tRNAs. As demonstrated previously, modifications as small as methylation inhibit endoribonucleolytic cleavage of tRNAs by angiogenin. What is the effect of N32 modifications (or modifications in general) on SLFN14 cleavage and its site-specificity? To address these questions, the authors should use a canonical (i.e., cytoplasmic) type I tRNA (e.g., yeast tRNA^{Phe} can be obtained commercially in its fully modified state). Yeast tRNA^{Phe} contains a 2'-O-methylated C32, which promotes the C3'-endo conformation found in canonical RNA duplexes. It would be interesting to see whether it interferes with efficient cleavage by SLFN14. Moreover, the authors should perform a cleavage experiment on modified and unmodified tRNA without using fluorescence-labels (using SYBR gold; judged from the gels shown in Fig. 3a-d, this should not result in an overly complex cleavage pattern). This should confirm cleavage in the anticodon and the absence of any acceptor-stem cleavage.

5. It is interesting to note the absence of 3'-fragments for cytoplasmic tRNA^{Ser} in Fig. 3d. The authors suspect this to be a cleavage event near the 3'-fluorophore. The authors should determine exactly where this cleavage occurs. After all, it may occur in the acceptor stem in line with the common cleavage pattern known from other SLFN family members. This could be achieved by doing complementary experiments using unlabeled tRNA^{Ser}.

6. The tRNA binding model proposed for anticodon-stem binding (Fig. 7a) appears to be significantly different from the one proposed for acceptor-stem binding by SLFN11 (PMCID: PMC9482658). a) Does their structure or sequence analysis provide any evidence for the molecular basis for this difference? Where does the specificity for the anticodon-stem-loop come from and why should SLFN14 not be unable to bind the tRNA acceptor stem in the same way as SLFN11? b) A major difference between the two models appears to be that the model proposed here supports a 2:2 stoichiometry of protein to tRNA, whereas the model by Metzner et al. (PMCID: PMC9482658) is compatible only with a 2:1 stoichiometry. To confirm their mode of binding, the authors could determine the stoichiometry of their complex, e.g., by using mass photometry.

Minor:

7. The authors claim that "Taken together, tRNA stem stability and solvent accessibility correlate with SLFN14 ribonuclease activity" (line 436). The former claim appears to be mainly based on the observation that two G:C pairs precede the cleavage sites in their two model substrates. Yet, this is not supported by any experimental data that would suggest that A:U pairs prevent cleavage.

8. For their model shown in Fig. 7, the authors use tRNA^{Sec}, because it contains consecutive G:C pairs in its anticodon stem and should thus be an efficient substrate for SLFN14 (see comment #7). However, this is not supported by the results shown in Suppl. Fig. 1d.

9. The authors suggest that the structural homology between the SLFN14 C-lobe and the DBD of SAMD9 supports the functional classification of SLFN14 as an anticodon nuclease (line 498). Since this homology should apply to other SLFN family members as well, does this imply that anticodon-cleavage is their original role and acceptor-stem cleavage is derived? Do the authors think that other SLFN proteins also cleave the anticodon?

Reviewer #3

(Remarks to the Author)

Van Riper et al. present a combined biochemical and structural analysis of human Schlafen 14 endoribonuclease. The authors found SLFN14 forms homodimers, cleaves several loop locations in tRNAs including the anticodon loop, contains a conserved ExxxxExK motif presumed to be responsible for catalysis, and characterized several disease-associated mutants. The authors report 5.6-6.5 angstrom resolution dimeric SLFN14 structures and modelled how the tRNA anticodon might bind and be cleaved by SLFN14. Overall, the text is clearly written and figures well presented. While the biochemical and biophysical analyses seem of reasonably good quality, the very low resolution of the structure renders the structural model overtly dependent on AlphaFold models instead of experimental data; and AlphaFold is known to be biased. There is also no tRNA-bound structure or footprinting analyses to inform substrate recognition, confirm the catalytic site, or ascertain the molecular mechanisms of disease-causing mutations. The tRNA-bound model is highly speculative as it is based on homology with a structure of SAMD9 bound to dsDNA, a very different substrate than tRNA. Altogether I think this work in its current form is quite preliminary and speculative, and does not provide sufficient amount of conceptual advance or definitive mechanistic insight to warrant publication in Nat Commun. I suggest the authors substantially improve upon the resolution of the structure if feasible and/or obtain additional experimental data about the tRNA or rRNA binding mode. Additional comments are below:

1. Line 426-437. The authors suggested that the G-C pairs are important for SLFN14 cleavage, but do not test it by mutating the pairs and showing reduced or blocked cleavage. I believe testing the effects of the G-C pairs directly would be important to support this conclusion.

2. Fig. 1f. why is the extent of tRNA cleavage so low, which then required contrast enhancement to visualize?

3. Fig. 1f and 1g. the predominant cleavage did not occur at the anticodon region, making it difficult to justify the classification of SLFN14 as primarily an anticodon nuclease, as implied by the title. Its activity towards rRNA seems much

higher than tRNAs. If it is a structure-specific endonuclease, how is it able to cleave rRNA to completion? Does the rRNA cleavage follow the observed tRNA patterns?

4. Fig. 2a legend, the authors should provide exact concentrations for each lane rather than a range.

5. Fig. 5c. 6b. These SDS-PAGE gels probably belong in the supplement than in main figures, as they don't provide much information.

6. Fig. 5f, why do the mutants bind better than WT and is there a super shift with the mutants?

7. Fig. 6d. Can the authors comment on why the delta mP max is changing so much while the affinity does not?

8. Fig. 7a. As mentioned above, I believe the structural model is too speculative as there are no distance constraints, footprints, structural data, or homology with a similar substrate.

9. Fig. 7b. the SLFN14 cartoon has not really changed in all three stages and provide little insight into the mechanisms of recognition or cleavage.

Version 1:

Reviewer comments:

Reviewer #1

(Remarks to the Author)

The authors have revised the manuscript, and the presented structural and biochemical data have been substantially improved. The manuscript now includes a 2.5 Å resolution cryo-EM reconstruction along with meaningful biochemical experiments.

Nevertheless, I have a few critical points regarding the presented biochemical data and the conclusions drawn about cleavage mechanism differences compared to published work. In general, the authors use relatively high protein concentrations of 200–400 nM SLFN14 and SLFN11 in their assays, which is two- to fourfold in excess of the RNA substrate concentration (100 nM). This might lead to artificial results and might be not appropriate for an enzymatic assay (e.g. SLFN11 assays)

Major Points:

Oligomeric State of SLFN14:

In Figure 1 and the results section (line 356), the authors state that the molecular weight of SLFN14 is 405 kDa, as determined by the gel filtration peak. This corresponds to something between a trimer and a tetramer. However, gel filtration is not a precise method for determining the oligomeric state of a protein. The authors cannot definitively describe SLFN14 as a discrete oligomer when its molecular weight does not clearly correspond to either a trimer or a tetramer, especially given that the cryo-EM structure indicates a dimer. The authors should confirm the oligomeric state in solution using mass photometry or another technique such as analytical ultracentrifugation or dynamic light scattering in different buffers. Alternatively, they should revise this part of the manuscript, including the discussion (line 686), to reflect the uncertainty.

Mg²⁺ vs. Mn²⁺ in Nuclease Assays:

In Figure 1d and most other nuclease assays, the authors use Mg²⁺ as the catalytic ion. However, Supplementary Figure 2c evaluates the influence of different metal ions on SLFN11 and SLFN14 nuclease activity. These data indicate that both Mg²⁺ and Mn²⁺ support comparable nuclease activity for SLFN14. In the case of SLFN11, Mn²⁺ leads to higher nuclease activity against tRNA^{T^{ser}}. The data suggests that SLFN11 also exhibits nuclease activity with Mg²⁺ at high enzyme concentrations, but its enzymatic activity is higher in the presence of Mn²⁺. The authors should perform the ribosomal RNA cleavage assay additionally with Mn²⁺ and compare the results to those with Mg²⁺.

Cryo-EM Density and Active Site Analysis:

The cryo-EM density does not clearly indicate whether one or two metal ions are bound in the active site. There appears to be unmodeled space in the density, which could accommodate a second metal ion or RNA. Given the flexibility and quality of the cryo-EM map in this region, modeling the RNA substrate and the active site is challenging. However, the current data do not support the conclusions drawn in Figures 5C and 6, as well as in lines 610, 648, and beyond. Similarly, the claim that residue orientations (E206 and D249) differ from those in other nucleases, including SLFN11, is not well supported by the density maps and models (Supplementary Figure 7f). The resolution does not allow for such definitive conclusions.

In all three cryo-EM maps, the first RNA base (chain D C1 and chain C G1) does not fit well within the cryo-EM density.

However, unmodeled density closer to the active site suggests potential differences in RNA binding. The authors should consider performing a masked refinement including only the N-terminal and RNA-binding regions, without imposing C2 symmetry, to account for possible differences between the two active sites.

Magnesium Coordination and Catalytic Mechanism:

Typically, Mg²⁺ is octahedrally coordinated, but in the SLFN14 structure, only three coordination sites are observed within a reasonable distance (2–4 Å). Given the missing model, additional unmodeled density in proximity, and the results in Supplementary Figure 2, the discussion on the Mg²⁺-dependent catalytic mechanism should be more cautious.

Minor Comments:

Line 630: "RNA catalysis" should be corrected to "RNA cleavage."

Supplementary Figures 1 and 2: The resolution in the PDF file is low and should be improved.

The manuscript has been substantially improved compared to the initial submission. However, several points should be either better discussed or validated using additional techniques.

Reviewer #2

(Remarks to the Author)

In their manuscript, Van Riper et al. report a functional and structural characterization of human SLFN14. Importantly, this manuscript is less a revision of the previous version submitted to NC but rather a new manuscript. Their conclusions are now based on an entirely new set of structural and functional data, leading to substantially different conclusions. Most importantly, their new data suggest that SLFN14 is not an anticodon-specific endonuclease, but rather exhibits specificity for the acceptor stem, very similar to other SLFN proteins. This new manuscript represents a significant improvement over the first version. The new enzymatic assays are well executed and now convincingly demonstrate the acceptor-stem specificity of SLFN14. Notably, they show that this specificity occurs only on native, presumably mature, tRNAs, whereas SLFN14 shows promiscuous cleavage on unmodified synthetic tRNAs. Unfortunately, the potential contribution of RNA modifications to this change in specificity remains unclear. One of the main issues raised previously by all three reviewers was the comparatively low resolution of the cryo-EM reconstruction. The authors now report a significantly improved cryo-EM reconstruction of SLFN14 to 2.5 Å, which even appears to have RNA bound in the active site cleft. Overall, this manuscript has significantly improved in data quality and clarity and will provide an important contribution to the field.

The authors should address the following points:

1. The map shows significant signs of orientation bias and oversharpening, especially in the NTDs of both subunits (e.g., bottom panel in Fig. 3a). This is also suggested by the angular distribution plots in Suppl. Fig. 6. Given that the interpretation on this region is critical for the main conclusions of this manuscript (e.g. including an inward-outward motion of the NTDs), it would be important either to solve the problem by collecting additional tilted data or rebalancing orientations, or at least to provide quantitative estimates for the orientation bias and include them into the discussion.
2. On p. 19 the authors state that "We did not observe asymmetric features and, therefore, applied C2 symmetry for the final consensus 3D reconstruction...". This is problematic, since the authors report an RNA bound structure of SLFN14, which, necessarily, introduces asymmetry into the active site. Thus, with RNA bound, the two orientations of the dimer cannot be identical, and using C2 symmetry necessarily introduces artifacts into the RNA density. Maybe the authors could perform masked refinement focussed on the NTDs to solve the symmetry issue? This may also improve the overall quality of the local reconstruction in this region.
3. On p. 20 the authors discuss the possible source/identity of the bound RNA and suggest it may be co-purified native nucleic acid. This could be tested by looking at the dataset that was collected without the addition of tRNA during sample preparation.
4. On p. 22 the authors state that "The SLFN14•RNA structure reveals a sequence-independent RNA binding surface along its N-lobes". Although sequence-independent electrostatic interactions indeed seem to make up much of this interface, the density for protein side chains in the interface and in particular the density for the RNA do not seem of sufficient resolution/quality to allow such an assertion.
5. On p. 26 the authors discuss the "Structural homology with prokaryotic restriction endonucleases". Do the authors claim this to be an actual "homology" between SLFN proteins and restriction endonucleases (i.e., due to descent from a common ancestor)? In this case they should demonstrate that this homology extends beyond the active site to the overall fold of both protein families. Alternatively, the authors should instead use the term "analogy" to describe convergent similarities in the active site architectures.

Reviewer #3

(Remarks to the Author)

Van Riper et al., substantially revamped their study on human SLFN14, and now reports a 2.5 angstrom resolution cryo-EM structure of SLFN14 bound to a double-stranded RNA helix, possibly as part of a bound tRNA acceptor stem. The authors have also re-examined the tRNA cleavage sites by SLFN14 and SLFN11, and found that SLFN14 cleavage sites can depend on tRNA modifications. These major changes, in particular the dramatic resolution improvement of their structures, have greatly improved the study. I believe the new conclusions reached in the revamped manuscript are now well supported by their experimental structure and associated data. I now support the publication of this work.

Version 2:

Reviewer comments:

Reviewer #1

(Remarks to the Author)

The authors have made significant improvements in response to reviewer feedback over the course of the revisions. The manuscript is now scientifically sound and clearly written. All previously raised concerns have been satisfactorily addressed.

Reviewer #2

(Remarks to the Author)

The authors have addressed all my concerns.

One minor issue is that the authors refer to the a Coulomb potential map as "electron density", which should be corrected.

Apart from that, I can recommend the manuscript for publication in Nature Communications

NCOMMS-23-46926 Rebuttal

Reviewer comments (black) and our response (blue).

Reviewer #1 (Remarks to the Author):

Van Riper et al. examined the cryo-EM structure and endonucleolytic activity of human endoribonuclease SLFN14 in their manuscript. SLFN14, a post-transcriptional regulator, plays a role in RNA degradation, particularly ribosomal RNA (rRNA), transfer RNA (tRNA), and messenger RNA (mRNA). Its nuclease activity has been associated with antiviral defense mechanisms and platelet function. Despite this, the molecular mechanisms regulating SLFN14 substrate recognition and catalytic activation have remained unclear. Dysregulation of SLFN14 is implicated in human diseases, including ribosomopathies and inherited thrombocytopenia, underscoring the need to understand its RNA processing activity and functional mechanism. Recently the Schlafen family of proteins has got attention among researchers seeking to understand the detailed mechanism of this conserved protein family. While X-ray and cryo-EM structures of human SLFN12, SLFN11, SLFN5, and partially SLFN13 have been solved, providing reasonably accurate AlphaFold predictions for all family members in monomeric and dimeric forms, human SLFN14 has been the least studied protein to date. Its biochemical and structural characterization represents an important advancement for the field.

We thank the reviewer for their support of our work and constructive comments, which have significantly improved our manuscript. Below, we address point-by-point the individual concerns of the reviewer.

Unfortunately, the resolution of the cryo-EM structure is relatively low, preventing detailed molecular conclusions. Conserved active site residues for all SLFN proteins have been proposed in earlier publications. From Supplementary Figure 4, it is unclear whether the authors attempted masked refinement of the N- and C-terminal domains separately to enhance the resolution of individual domains. Could additional data collection enhance the resolution of the structure?

The revised manuscript presents a new 2.5 Å resolution cryoEM reconstruction obtained from data collected on a Krios G4 microscope. Following extensive image processing, we now observe side chain density across all the SLFN14 domains, including its nuclease active site. Furthermore, we observed additional density within the RNA cleft that has the characteristic features of duplexed nucleic acid. This improvement of the cryoEM data strengthens the conclusions of this study and provides new insight into SLFN14 RNA binding and cleavage.

While it is assumed that the active site of SLFN14 largely resembles that of other SLFNs e.g. SLFN11 and 12, the paper lacks a direct structural comparison and overlay of the active sites. This comparison could shed light on the similarities and differences in substrate recognition, especially between SLFN11 and SLFN14. The structure of SLFN11 complexed with tRNA has been solved. Did the authors attempt to solve complex structures with the identified SLFN14 substrates, or is that feasible within the revision timeframe?

Thank you for this suggestion. We now provide a structural comparison and overlay of the SLFN14, SLFN13, SLFN12, SLFN11, and SLFN5 active site residues (see Figure 5c and Supplementary Figure 14 in the revised manuscript). Moreover, we collected new cryoEM data in the presence of native tRNA with the intent of revealing RNA-protein interface(s). In this new cryoEM reconstruction, we observed the density of nucleic acid within the RNA cleft, revealing a nucleic acid interface at the SLFN14 N-lobes. We discuss the similarities between the SLFN14 sequence-independent RNA interface observed in our structure and the N-lobe tRNA binding sites of SLFN11, suggesting a shared mechanism of RNA binding.

SLFN14's ribonuclease activity depends on magnesium ions in contrast to SLFN11, which is solely active with manganese. Could the authors test SLFN14's ability to cleave substrates with other metal ions, such as manganese, and determine if different ions lead to distinct cleavage products?

The revised manuscript now includes Supplementary Figure 2, which characterizes tRNA cleavage activity by SLFN14 and SLFN11 in the presence of different metal ions. We show that 1 mM Mg^{2+} can support both SLFN14 and SLFN11 tRNA cleavage activity in our *in vitro* study. The addition of 1 mM Mn^{2+} can also support SLFN14 and SLFN11 nuclease activity, although the appearance of short cleavage products suggests the possibility of nonspecific nuclease activity. Conversely, Metzner et al. (2022) detected SLFN11 tRNA cleavage activity when the 2 mM Mg^{2+} -containing reaction buffer was supplemented with 2 mM Mn^{2+} . This discrepancy in metal dependency could be due to differences in SLFN11 host expression systems, tRNA sequence, or divalent metal ion concentration.

The high protein concentration used in all cleavage experiments, as evident in e.g. Figure 3, suggests SLFN14 may be a relatively unproductive enzyme. Could the authors please explain this behavior?

We agree that the SLFN14 tRNA cleavage activity is relatively less productive than its SLFN11 counterpart. We have now highlighted this point in the text and added SLFN11 tRNA cleavage reactions for reference in revised Figure 2d and Supplementary Figure 4. We suspect that SLFN14 may have a preferred target that is not a tRNA or that our *in vitro* system may lack an important signal to stimulate its nuclease activity. Indeed, the efficiency of the SLFN11 and SLFN12 tRNA nucleases are modulated by external signals, such as protein phosphorylation and binding partners (Kugler et al. (2024), Yan et al. (2022), Garvie et al. (2021)). Whether SLFN14 encodes a similar regulatory phosphorylation site and/or relies on regulatory binding partners, such as the ribosome, is discussed in the revised manuscript.

While the nuclease assays in Figures 1 and 3 are convincing and include necessary controls, the fragmented tRNA experiment in Figure 2c-d is puzzling. The tRNA cartoon is a simplified representation, and dividing tRNA into minimal pieces lead to a different folding of the individual parts. Hence this experiment may lead to incorrect assumptions. Therefore, it is advisable to remove the entire experiment. Could the authors consider sequencing the tRNA cleavage products instead?

We have removed the fragmented tRNA experiment from the revised manuscript. Instead, we performed a side-by-side tRNA cleavage study with the well-characterized SLFN11 family member (see revised Figure 2). We demonstrate that, while SLFN14 exhibits broader site-specificity with unmodified synthetic tRNA, both SLFN14 and SLFN11 cleave modified native tRNA similarly, resulting in indistinguishable RNA fragments that correspond to the well-established acceptor stem cleavage site of SLFN11.

In Supplementary Figure 2, the authors demonstrate SLFN14's ability to bind single-stranded DNA (ssDNA). In SLFN11, ssDNA binds on top of the C-terminal helicase domain. Could the authors use mutational or structural studies to pinpoint where SLFN14 binds ssDNA?

SLFN14 does not encode the equivalent basic residues observed within the ssDNA binding site of the SLFN11 CTD, suggesting this site may be a unique feature of SLFN11. We suspect the low-affinity ssDNA and dsDNA binding properties of SLFN14 observed *in vitro* may be nonspecific binding at the RNA cleft. In the absence of a clear mechanism for this molecular interaction or an explanation of its relevance, we decided to remove this preliminary *in vitro* experiment from the revised manuscript.

The presented data lead to a new tRNA binding model in Figure 7, differing from the SLFN11 tRNA binding mode. A detailed comparison of the residues involved in substrate recognition in both proteins would provide insights into the differences among the structurally conserved but functionally diverse Schlafen family members. Based on new structural and biochemical data presented in the revised manuscript, we now provide a revised RNA binding model that closely aligns with the existing SLFN11 tRNA binding mode (Metzner et. al. (2022), Kugler et. al. (2024)). Our biochemical study reveals that, unlike SLFN11, SLFN14 depends on modified native tRNA for site-specificity at the acceptor stem. Our structural analysis with SLFN11 suggests that SLFN14 likely binds the T-arm through its N-lobes and the acceptor stem within a neighboring C-lobe (see revised Figure 4, Supplementary Figure 12 and Supplementary Figure 13).

In summary, the findings are novel and significant in the field of the Schlafen family of proteins. While the work provides valuable insights, enhancing the structure's quality or presenting complex structures of SLFN14 with tRNA substrates would strengthen the validity of the proposed model.

We thank the reviewer for their support of our manuscript. Our revised manuscript now includes a high-resolution structure of RNA-bound SLFN14 along with additional biochemical studies that together allow us to revise and strengthen our proposed model for SLFN14-direct tRNA binding and cleavage.

Minor comments:

In Figure 3e, indicating the labels 3'fluorescein and 5'Cy5 in the tRNA cartoon would help the reader.

We thank the reviewer for their suggestion. To eliminate the possibility of an artifact from the bulky tRNA fluorophores, we removed this cartoon along with its corresponding

fluorescently labeled tRNA experiments. In the revised manuscript, we characterize SLFN14 nuclease activity using unlabeled synthetic and native RNA substrates.

It was not possible to play Video 3.

Our apologies. We have recreated all our movies with the mp4 extension to avoid issues with viewing our movie files.

Reviewer #2 (Remarks to the Author):

In their manuscript, Van Riper et al. perform a functional and structural characterization of human SLFN14, a member of the Schlafen (SLFN) family of endoribonucleases. Using recombinantly expressed and purified full-length protein, the authors report a cryo-EM reconstruction of the SLFN14 at 5.5 Å resolution, which reveals a homodimeric architecture and conformational flexibility. Moreover, the authors perform an in vitro biochemical characterization of SLFN14, demonstrating a magnesium-dependent ribonuclease activity, depending on an ExxxxExK catalytic motif, consistent with previous analyses of other SLFN homologs. Notably, these analyses reveal the ability of SLFN14 to cleave tRNAs in the anticodon loop rather than the acceptor stem, which leads the authors to propose that SLFN14, in notable contrast to other previously characterized SLFN homologs, is an anticodon-stem-loop-specific endoribonuclease.

Overall, this work is a timely and potentially important contribution to the field of tRNA-cleaving ribonucleases and their diverse roles in gene-regulatory circuits. In particular, the proposed mechanism of anticodon-specific cleavage by SLFN14, if true, would be a critical novel insight. The experimental work is well-executed. A minor caveat is the relatively low resolution of 5.5 Å of the reported cryo-EM reconstruction, which, however, appears to be of sufficiently high quality to support the main claims regarding the structure made by the authors.

We thank the reviewer for their support of our work and appreciate their valuable feedback. Please see below for our point-by-point responses.

One major concern is insufficient reference to earlier work, both regarding the mechanistic basis for SLFN-mediated RNA cleavage in general, but in particular earlier structural work. Metzner et al. previously reported a 2.86 Å reconstruction for the human SLFN11 homodimer (PMCID: PMC9482658), which appears to be highly similar to the one reported here for human SLFN14. Although a direct structural comparison between the two SLFN homologs (and the structure of other homologs) would be highly relevant, such a comparison is missing completely.

We agree with the reviewer and have added additional references of earlier work to the revised manuscript to strengthen our structural and biochemical comparison of SLFN14 with other members. This includes references to existing structural work of the human SLFN11 homodimer bound to tRNA (e.g. Metzner et al. (2022); Kugler et al. (2024)). The revised manuscript reports a high degree of overlap between the tRNA-bound SLFN11 structures and the new SLFN14-RNA cryoEM structure reported in this work, suggesting a shared mechanism of T-arm recognition by the SLFN N-lobes (see revised

Supplementary Figure 12). Revised Figure 5 and Supplementary Figure 14 also describe similarities and differences in the active site arrangement across the SLFN family. Finally, we report the structural homology of the SLFN14 disease hotspot associated with inherited thrombocytopenia and SLFN11 basic residues contributing to tRNA binding interfaces (revised Supplementary Figure 13). Together, this analysis suggests a potential dysregulation in RNA recruitment or binding in SLFN14-linked inherited thrombocytopenia patients.

Given that the authors propose a radically new mode of tRNA binding and cleavage, such a comparison would be informative as to how these two closely related enzymes achieve such distinct substrate recognition and cleavage mechanisms.

In the revised manuscript, we perform *in vitro* RNA cleavage reactions with unmodified synthetic tRNA to confirm earlier work (e.g. Metzner et. al. (2022)) that recombinant SLFN11 cleaves the acceptor stem of unmodified synthetic tRNA, whereas SLFN14 often cleaves outside of the acceptor stem. SLFN14 tRNA cleavage activity is also relatively unproductive compared to SLFN11's catalytic output, suggesting a difference in SLFN14's substrate preference or a lack of an important stimulatory signal. We repeated our *in vitro* RNA cleavage assays with native tRNA to determine whether intrinsic tRNA features, such as modifications, can influence SLFN site-specificity. In this new study, we demonstrate that SLFN14, but not SLFN11, typically depends on modified native tRNA for site-specificity of the acceptor stem (see revised Figure 2 and Supplementary Figure 4). While the underlying molecular basis for this distinction in site-specificity between SLFN14 and SLFN11 remains unclear, we highlight differences in their active site organization and discuss possible mechanisms for modification-dependent tRNA processing in the discussion section.

Also, the authors assert some observations in a manner that they appear novel. For example, in lines 310-313 the authors write: "Multiple sequence alignments across the SLFN family identify a consensus ExxxxExK (where x is any residue) motif harboring two invariant glutamates and a well-conserved lysine residue (Fig. 1b). We define this discrete element as a putative catalytic motif...". This is not an original observation, but has been noted previously (e.g., PMID: PMC5862951, PMC9256597, PMC9482658). The authors should take care to mention the previous work where appropriate.

Thank you and we agree. We have revised the entire text to include references to highlight important discoveries from earlier work, including observations related to the catalytic residues of the SLFN family. Specifically, we reference PMID: PMC5862951, PMC9256597, PMC9482658, and others when introducing the catalytic motif and performing comparative analysis.

The second major concern regards the claim that SLFN14, in stark contrast to all other previously characterized SLFN homologs, is anticodon-stem-loop specific: "Our work shows that SLFN14 is not an acceptor stem nuclease but an anticodon stem-loop nuclease" (line 617). Thus, the authors specifically exclude the possibility that SLFN14 cleaves the acceptor-stem at all. This claim is based exclusively on *in vitro* cleavage assays with synthetic, unmodified RNAs, labeled on one or both ends with bulky fluorophores. Below, I elaborate on several specific points.

In the revised manuscript, we remove all experiments with fluorescently labeled RNA substrates to replace these studies with unlabeled, unmodified synthetic and modified native substrates. As the reviewer anticipated, this new analysis demonstrates that SLFN14's broad site-specificity against unmodified synthetic tRNA is altered in the presence of modified native tRNA, where SLFN14 exclusively cleaves the acceptor stem of native targets. Please see revised Figure 2 and Supplementary Figure 4.

1. The analysis starts with cleavage assays on small RNAs isolated from human cell lines (Fig. 1f and Suppl. Fig 1b). These indeed show formation of short, ca. 30 nt long fragments; however, the bands with the highest intensity correspond to ca. 65 nt. This would agree with a cleavage position at the base of the acceptor stem of type I tRNAs (corresponding fragments from type II tRNAs would run together with full-length type I tRNAs). The Northern blot results in Suppl. Fig. 1d appear to further confirm this, showing strong bands immediately below the full-length tRNAs. No bands are visible for tRNA-halves. The authors should explain a) how the longer fragments are generated if not by acceptor-stem cleavage, and b) why there are no ~30 nt bands in the northern blots in Suppl. Fig. 1. c) If this is due to the characteristics of the probes used for northern blotting, the authors should repeat the experiment at least for some of the tRNAs with probes complementary to the 30 nt 5'-fragments. This will be important to demonstrate that SLFN14 not only cleaves the anticodon on unmodified synthetic tRNAs, but also on mature and fully modified tRNAs. d) Is there a rationale as to why the authors focus exclusively on the small fragments (~30 nt) and completely disregard the longer (~65 nt) fragments in their biochemical characterization?

We thank the reviewer for their constructive comment. To address this concern, we performed a follow-up study to compare SLFN14-derived tRNA products from a subset of type-I and type-II unmodified synthetic and modified native tRNAs. We visualized the RNA fragments by SYBR gold and Northern blot analysis. This new characterization demonstrates that SLFN14 often cleaves outside of the acceptor stem with unmodified synthetic tRNA, but exclusively cuts the acceptor stem of modified native tRNA (see revised Figure 2 and Supplementary Figure 4). The revised manuscript underscores the importance of SLFN14 acceptor stem cleavage and discusses the possible molecular mechanisms that could explain modification-dependent tRNA processing.

2. The rationale to use mitochondrial tRNAMet as model substrate remains unclear. Given that the authors claim to demonstrate “an essential role for the tRNA elbow, a universal structural feature formed through tertiary interactions between the D-loop and T-loop” (line 381), it is important to note that mt-tRNAMet does not contain a canonical elbow and may not form any of those D-loop-T-loop tertiary interactions that define the classical cytosolic tRNA elbow. Moreover, unless SLFN14 enters into the mitochondria, its cleavage pattern of mt-tRNAMet is unlikely to be physiologically relevant at all. Thus, unless there is a specific reason to use it, mt-tRNAMet is a poor choice as a model substrate for SLFN14 (the reference to its use by Metzner et al (PMCID: PMC9482658) appears unmotivated). The authors should use a more likely and thus physiological relevant substrate, such as a eukaryotic cytoplasmic type I tRNA (see below).

We agree. We have removed the mitochondrial tRNA substrate and now use cytosolic type-I and type-II tRNAs in our revised studies.

3. To demonstrate dependency on the tRNA elbow, the authors perform cleavage assays with 5'Cy5-labeled mt-tRNAMet fragments of different length (Fig. 2c-d). They conclude that cleavage depends on the elbow (line 381), albeit indirectly (line 640). However, their binding model (Fig. 7a) still begs the question of how this indirect effect comes about and why the anticodon-stem-loop substrate is not cleaved at all (Fig. 2c-d). Moreover, they conclude that the acceptor stem is dispensable (line 377). This, of course, is only true for anticodon-cleavage, which they focus on. It does not demonstrate that SLFN14 does not interact with the acceptor stem in a more SLFN11-like manner and may even cleave it on unlabeled or/and modified substrates. To demonstrate this, the authors should do cleavage and binding assays with the same tRNA fragments (or preferably with those from a cognate substrate instead of the mt-tRNA) in the absence of fluorophores (or repeat the same pattern with 3'-label). We removed all experiments using mitochondrial tRNA, including the experiment referenced in this point. In the revised manuscript, we rely solely on RNA cleavage assays using unlabeled cytosolic tRNA. Our new data suggests that SLFN14 site-specificity is different in the absence (synthetic tRNA) and presence (native tRNA) of modifications. Strikingly, SLFN14 exclusively cleaves the acceptor stem of native tRNAs, implying a common mechanism of tRNA cleavage to other SLFN family members (see revised Figure 2 and Supplementary Figure 4). Our new high-resolution cryoEM structure of SLFN14 also reveals the density of nucleic acid in its RNA cleft. Structural analysis with existing tRNA-bound SLFN11 structures (e.g. Metzner et. al. (2022), Kugler et. al. (2024)) suggests a common mode of T-arm recognition for tRNA binding (revised Supplementary Figure 12).

4. A central problem is that the claim of anticodon-specificity is based entirely on *in vitro* cleavage assays using chemically synthesized tRNAs with one or two bulky fluorophores on their 5' and 3' ends.

a) These fluorophores may interfere with productive acceptor stem binding by SLFN14, thereby promoting a (possibly otherwise less-favored) cleavage in the anticodon.

To address this concern, we repeated our *in vitro* RNA cleavage reactions in the absence of fluorophores and showed that the fluorophore can influence SLFN14 RNA processing. Therefore, we have replaced all fluorescently labeled RNA substrates with unlabeled RNAs in the revised manuscript. Using unlabeled RNA, we now demonstrate spurious SLFN14 nuclease activity against unmodified synthetic tRNA, but site-specific acceptor stem cleavage of modified native tRNA.

b) Position 32 in the anticodon-loop is modified in the majority of cytosolic tRNAs. As demonstrated previously, modifications as small as methylation inhibit endoribonucleolytic cleavage of tRNAs by angiogenin. What is the effect of N32 modifications (or modifications in general) on SLFN14 cleavage and its site-specificity? To address these questions, the authors should use a canonical (i.e., cytoplasmic) type I tRNA (e.g., yeast tRNAPhe can be obtained commercially in its fully modified state). Yeast tRNAPhe contains a 2'O-methylated C32, which promotes the C3'-endo conformation found in canonical RNA duplexes. It would be interesting to see whether it interferes with efficient cleavage by SLFN14. Moreover, the authors should perform a

cleavage experiment on modified and unmodified tRNA without using fluorescence-labels (using SYBR gold; judged from the gels shown in Fig. 3a-d, this should not result in an overly complex cleavage pattern). This should confirm cleavage in the anticodon and the absence of any acceptor-stem cleavage.

We thank the reviewer for their suggestion. Unfortunately, the commercial yeast tRNAPhe has been discontinued. Instead, we performed SLFN14 RNA cleavage assays with unlabeled synthetic (unmodified) and unlabeled native (presumably modified) tRNAs. SLFN14 often cleaves outside of the acceptor stem in the absence of modifications but exhibits remarkable acceptor stem site-specificity in the presence of modifications. This modification-dependent site-specificity was not observed with SLFN11 in the conditions tested. We highlight these important observations in the revised manuscript and discuss the possible molecular basis for modification dependency (e.g. the precedence for tRNA modifications shielding nuclease processing sites) in the revised discussion.

5. It is interesting to note the absence of 3'-fragments for cytoplasmic tRNASer in Fig. 3d. The authors suspect this to be a cleavage event near the 3'-fluorophore. The authors should determine exactly where this cleavage occurs. After all, it may occur in the acceptor stem in line with the common cleavage pattern known from other SLFN family members. This could be achieved by doing complementary experiments using unlabeled tRNASer.

As suggested by the reviewer, we performed follow-up experiments with unlabeled serine tRNA. This additional work led us to believe that the bulky fluorophores were indeed influencing the SLFN14 RNA cleavage pattern. For this reason, we removed all the fluorescently labeled RNA experiments and now solely report the RNA cleavage activity of unlabeled RNA visualized either by SYBR gold or Northern blot analysis. This revised analysis indicates that SLFN14 processing specificity towards the acceptor stem is modification-dependent. We explain the possible reasons for this observation in the revised discussion.

6. The tRNA binding model proposed for anticodon-stem binding (Fig. 7a) appears to be significantly different from the one proposed for acceptor-stem binding by SLFN11 (PMCID: PMC9482658). a) Does their structure or sequence analysis provide any evidence for the molecular basis for this difference? Where does the specificity for the anticodon-stem-loop come from and why should SLFN14 not be unable to bind the tRNA acceptor stem in the same way as SLFN11? b) A major difference between the two models appears to be that the model proposed here supports a 2:2 stoichiometry of protein to tRNA, whereas the model by Metzner et al. (PMCID: PMC9482658) is compatible only with a 2:1 stoichiometry. To confirm their mode of binding, the authors could determine the stoichiometry of their complex, e.g., by using mass photometry. We removed the original tRNA binding model (originally Figure 7a) because we have new data indicating that modified native tRNAs are cleaved within the acceptor stem (see revised Figure 2). Based on this new evidence, we propose that SLFN14 follows an SLFN11-like tRNA binding mode of 2:1 stoichiometry (SLFN14:tRNA). This is further supported by our new high-resolution RNA-bound SLFN14 cryoEM structure and accompanying structural comparison with the tRNA-bound SLFN11 structures (Kugler

et al. (2024)). We demonstrate in the revised Supplementary Figures 12 and 13 a high degree of overlap between the position of the phosphodiester backbone of their respective RNA and the placement of equivalent basic residues contributing to SLFN11 tRNA interfaces.

Minor:

7. The authors claim that “Taken together, tRNA stem stability and solvent accessibility correlate with SLFN14 ribonuclease activity” (line 436). The former claim appears to be mainly based on the observation that two G:C pairs precede the cleavage sites in their two model substrates. Yet, this is not supported by any experimental data that would suggest that A:U pairs prevent cleavage.

We thank the reviewer for their suggestion. We have removed this statement from the revised manuscript due to the inhibition of anticodon stem cleavage by SLFN14 in the presence of modified native tRNA. The revised manuscript focuses on acceptor stem cleavage by SLFN14 because this is the primary processing site within native tRNAs.

8. For their model shown in Fig. 7, the authors use tRNAs^{SeC}, because it contains consecutive G:C pairs in its anticodon stem and should thus be an efficient substrate for SLFN14 (see comment #7). However, this is not supported by the results shown in Suppl. Fig. 1d.

We have removed this model from the revised manuscript. Our new biochemical characterization of SLFN14 processing demonstrates a strong preference for acceptor stem cleavage within native tRNAs. However, SLFN14 processing of native tRNAs^{SeC} demonstrates an atypical processing pattern that does not align with acceptor stem cleavage. We added a statement to the results section that this noncanonical cleavage event may be explained by the unusual secondary and tertiary structure of tRNAs^{SeC}.

9. The authors suggest that the structural homology between the SLFN14 C-lobe and the DBD of SAMD9 supports the functional classification of SLFN14 as an anticodon nuclease (line 498). Since this homology should apply to other SLFN family members as well, does this imply that anticodon-cleavage is their original role and acceptor-stem cleavage is derived? Do the authors think that other SLFN proteins also cleave the anticodon?

As described in the responses above, the revised manuscript clarifies that SLFN14 only exhibits site-specificity towards the anticodon arm in the absence of modifications. Unlike SLFN14, SLFN11's acceptor stem specificity is not dependent on modifications in the conditions tested. For this reason, we removed the statement that SLFN14 can be classified as an anticodon nuclease.

Reviewer #3 (Remarks to the Author):

Van Riper et al. present a combined biochemical and structural analysis of human Schlafen 14 endoribonuclease. The authors found SLFN14 forms homodimers, cleaves several loop locations in tRNAs including the anticodon loop, contains a conserved ExxxxExK motif presumed to be responsible for catalysis, and characterized several disease-associated mutants. The authors report 5.6-6.5 angstrom resolution dimeric

SLFN14 structures and modelled how the tRNA anticodon might bind and be cleaved by SLFN14. Overall, the text is clearly written and figures well presented. While the biochemical and biophysical analyses seem of reasonably good quality, the very low resolution of the structure renders the structural model overtly dependent on AlphaFold models instead of experimental data; and AlphaFold is known to be biased. There is also no tRNA-bound structure or footprinting analyses to inform substrate recognition, confirm the catalytic site, or ascertain the molecular mechanisms of disease-causing mutations. The tRNA-bound model is highly speculative as it is based on homology with a structure of SAMD9 bound to dsDNA, a very different substrate than tRNA. Altogether I think this work in its current form is quite preliminary and speculative, and does not provide sufficient amount of conceptual advance or definitive mechanistic insight to warrant publication in Nat Commun. I suggest the authors substantially improve upon the resolution of the structure if feasible and/or obtain additional experimental data about the tRNA or rRNA binding mode. Additional comments are below:

We thank the reviewer for their valuable feedback, which has greatly improved the strength of the conclusions presented in the revised manuscript. In summary, we present a new cryoEM structure of RNA-bound SLFN14 at 2.5 Å resolution, perform additional structural analysis to propose an SLFN11-like tRNA binding mode, and demonstrate SLFN14's dependency on native tRNA for acceptor stem site-specificity. We replaced the original tRNA-bound model guided by SAMD9 homology with the analysis of our cryoEM-derived structure, showing a sequence-independent RNA binding surface at the N-lobes of the dimeric RNA cleft. A comparative study with existing tRNA-bound SLFN11 structures (e.g. Kugler et al. (2024)) shows a high degree of overlap between their RNA clefts and the phosphodiester backbone of their respective RNA. Most notably, the arrangement of the duplexed RNA bound at the SLFN14 N-lobes is very similar to the T-arm interfaces seen in type-I and type-II bound SLFN11 structures (revised Figure 4, Supplementary Figure 12). Moreover, we show the placement of SLFN14 basic residues associated with inherited thrombocytopenia at the entrance of the RNA cleft, where equivalent basic residues in SLFN11 form interfaces with the acceptor stem, T-arm, and variable arm of tRNA (revised Figure 4, Supplementary Figure 13). The revised manuscript solely relies on unlabeled, unmodified synthetic and modified native tRNAs for the characterization of SLFN14 processing. This new biochemical study reveals the broad site-specificity of SLFN14 towards unmodified tRNAs and exclusive site-specificity for the acceptor stem of modified native tRNAs (revised Figure 2, Supplementary Figure 4). Unlike SLFN14, we show that SLFN11's acceptor stem specificity is not dependent on modifications in the conditions tested. Moreover, SLFN14 has a relatively unproductive tRNA cleavage activity compared to its SLFN11 family member, suggesting a difference in substrate preference or the lack of a stimulatory signal in our *in vitro* assay. We discuss potential molecular signals that could enhance the ribosome-associated SLFN14 nuclease, including the recent report that the unproductive Angiogenin tRNA nuclease relies on its direct association with the ribosome for catalytic stimulation and specificity (Loveland et al. (2024)).

1. Line 426-437. The authors suggested that the G-C pairs are important for SLFN14 cleavage, but do not test it by mutating the pairs and showing reduced or blocked

cleavage. I believe testing the effects of the G-C pairs directly would be important to support this conclusion.

We thank the reviewer for this suggestion. However, we have removed this statement from the revised manuscript based on new data presented in this work. We extended our biochemical characterization of SLFN14 to compare its site-specificity between unmodified and modified tRNAs. While SLFN14 often cleaves outside of the acceptor stem of unmodified tRNAs (e.g. downstream of G-C pairs within the anticodon arm), it exhibits strong site-specificity for the acceptor stem of native tRNAs tested in this study (see revised Figure 2). For this reason, we focus on the favored acceptor stem processing events observed with native tRNAs.

2. Fig. 1f. why is the extent of tRNA cleavage so low, which then required contrast enhancement to visualize?

Based on new biochemical data presented in the revised manuscript, we demonstrate that SLFN14 site-specificity is altered in the presence of tRNA modifications. While SLFN14 demonstrates robust processing of the anticodon arm when incubated with unmodified tRNA, it exhibits altered site-specificity with modified native tRNA, favoring cleavage of the acceptor stem. This is in good agreement with the original data presented in Fig. 1f, suggesting SLFN14 has limited site-specificity to the anticodon arm of native tRNAs, requiring contrast enhancement for their visualization. While this original data suggests there may be a low abundant population of native tRNAs that can be cleaved at the anticodon arm by SLFN14, we removed this preliminary *in vitro* data from the revised manuscript to focus on the prominent processing site of the acceptor stem.

3. Fig. 1f and 1g. the predominant cleavage did not occur at the anticodon region, making it difficult to justify the classification of SLFN14 as primarily an anticodon nuclease, as implied by the title. Its activity towards rRNA seems much higher than tRNAs. If it is a structure-specific endonuclease, how is it able to cleave rRNA to completion? Does the rRNA cleavage follow the observed tRNA patterns?

Based on the new biochemical analysis presented in the revised manuscript, we removed the classification that SLFN14 is an anticodon stem nuclease. As described in response to concern #2, we demonstrate that SLFN14 site-specificity is altered in the presence of modified native tRNAs, resulting in a preference for acceptor stem processing (revised Figure 2, Supplementary Figure 4). We also perform a side-by-side comparison of SLFN14 and SLFN11 tRNA processing to demonstrate that SLFN14's catalytic activity is relatively inefficient compared to its family member. This distinction may be due to a difference in SLFN14's preferred RNA target or the lack of an important stimulatory signal. We revised the text to include a discussion on the potential regulatory mechanisms that may influence SLFN14's catalytic output. Although it is beyond the scope of this study, future research will be important for identifying the physiological targets (e.g. tRNA, rRNA of mature ribosome particles, etc.) of SLFN14 during stress-induced translational control.

4. Fig. 2a legend, the authors should provide exact concentrations for each lane rather than a range.

We have now added the exact concentrations for each lane in the figure legends.

5. Fig. 5c. 6b. These SDS-PAGE gels probably belong in the supplement than in main figures, as they don't provide much information.

The original Figure 5c SDS-PAGE image was moved to the revised supplementary document (now Supplementary Figure 15a), and Figure 6c and its associated *in vitro* assay were removed due to its preliminary nature.

6. Fig. 5f, why do the mutants bind better than WT and is there a super shift with the mutants?

The data presented in the original Fig. 5f panel was an electrophoretic mobility shift assay demonstrating the RNA binding activity of SLFN14 active site variants using unmodified synthetic tRNA. In the revised manuscript, we demonstrate that SLFN14 site-specificity is altered in the presence of modified native tRNA, favoring the canonical SLFN acceptor stem processing site. For this reason, we replaced the original experiment with an SLFN14 RNA binding assay using modified native tRNAs (now Figure 5e). While the conclusion that SLFN14 active site variants retain RNA binding activity remains generally true, the super shift observed in the original experiment no longer appears specific to SLFN14 mutants in the revised assay. Interestingly, the SLFN14 R133K variant appears to form a most stable interaction with native tRNA compared to wild-type protein. We suspect this active site change may result in a subtle change in the electrostatic landscape or active site conformation.

7. Fig. 6d. Can the authors comment on why the delta mP max is changing so much while the affinity does not?

We thank the reviewer for their keen observation. While the data presented in the original Fig. 6d panel is interesting, we decided to remove this preliminary data from the revised manuscript since it relies on unmodified synthetic tRNA (see response to concern #6 for more details). In its place, we provide a structural comparison of the SLFN14 disease-associated residues observed in our high-resolution cryoEM structure, showing a high degree of similarity with equivalent SLFN11 tRNA interface residues (Kugler et. al. (2024)). See revised Supplementary Figure 13.

8. Fig. 7a. As mentioned above, I believe the structural model is too speculative as there are no distance constraints, footprints, structural data, or homology with a similar substrate.

We agree and have removed this model from the revised manuscript. Instead, we provide a structural comparison that relies on our new high-resolution RNA-bound SLFN14 cryoEM structure and recent tRNA-bound SLFN11 structures (Kugler et. al. (2024)). This analysis supports a working model where the L-shaped tRNA docks into the SLFN14 RNA cleft in an upright position. This SLFN11-like tRNA binding mode correlates well with the SLFN14 RNA binding interfaces observed at the N-lobes of our structure and aligns closely with our new biochemical studies showing the preferred acceptor stem processing site of native tRNA.

9. Fig. 7b. the SLFN14 cartoon has not really changed in all three stages and provide little insight into the mechanisms of recognition or cleavage.
We removed this cartoon from the revised manuscript.

NCOMMS-23-46926-Z Rebuttal

Reviewer comments (black) and our response (blue).

REVIEWER COMMENTS

Reviewer #1 (Remarks to the Author):

The authors have revised the manuscript, and the presented structural and biochemical data have been substantially improved. The manuscript now includes a 2.5 Å resolution cryo-EM reconstruction along with meaningful biochemical experiments.

We thank the reviewer for their support of our revised work. Please see below our response to their valuable feedback.

Nevertheless, I have a few critical points regarding the presented biochemical data and the conclusions drawn about cleavage mechanism differences compared to published work. In general, the authors use relatively high protein concentrations of 200–400 nM SLFN14 and SLFN11 in their assays, which is two- to fourfold in excess of the RNA substrate concentration (100 nM). This might lead to artificial results and might be not appropriate for an enzymatic assay (e.g. SLFN11 assays)

Thank you for this point. While the upper limit of the SLFN14/SLFN11 protein titration is in excess of substrate, the lower end is sub-stoichiometric and equimolar to RNA. The cleavage patterns observed at equimolar ratio is comparable to those seen with excess protein, although the fragment bands appear more intense with elevated protein concentration. Differences in activity level between our study and others may be due to differences in the reaction conditions. For example, our recombinant protein samples are sourced from mammalian cells which has a characteristic absorbance at 260 nm, suggesting the protein may co-purify with nucleic acid. This may indicate a mixed population of RNA-free and RNA-bound SLFN14/SLFN11 protein states. This is reminiscent to an earlier biochemical and structural study demonstrating SLFN11 can remain stably bound to tRNA products, limiting its tRNA cleavage activity in vitro (Kugler et al. 2024). The revised manuscript states the possibility that the SLFN14 sample may co-purify with cellular-sourced nucleic acid. Please see “Architecture of the dimeric SLFN14 ribonuclease” in the revised results section.

Major Points:

Oligomeric State of SLFN14:

In Figure 1 and the results section (line 356), the authors state that the molecular weight of SLFN14 is 405 kDa, as determined by the gel filtration peak. This corresponds to something between a trimer and a tetramer. However, gel filtration is not a precise method for determining the oligomeric state of a protein. The authors cannot definitively describe SLFN14 as a discrete oligomer when its molecular weight does not clearly correspond to either a trimer or a tetramer, especially given that the cryo-EM structure indicates a dimer. The authors should confirm the oligomeric state in solution using mass photometry or another technique such as analytical ultracentrifugation or dynamic light scattering in different buffers. Alternatively, they should revise this part of the manuscript, including the discussion (line 686), to reflect the uncertainty.

We agree. We have revised the statement on line 356 to “*SLFN14 resolves as a single homogenous peak suggesting SLFN14 is stable in solution.*” To adequately determine the discrete oligomeric state of SLFN14, we present mass photometry data in the revised manuscript (please see Supplementary Figure 2). In storage buffer containing 150 mM NaCl, we observe a prominent peak at ~232 kDa and a minor peak at ~456 kDa, corresponding with an SLFN14 homodimer and homotetramer, respectively. In high salt conditions, we observe a minor peak at ~105 kDa and a prominent peak at ~212 kDa, representing an SLFN14 monomer and homodimer, respectively. While the homotetramer is salt-sensitive, the SLFN14 homodimer is salt-resistant. This is a notable distinction to SLFN11 which forms a salt-sensitive monomer-dimer equilibrium (Metzner et al. 2022). We suspect that the hydrophobic nature of the SLFN14 dimerization interface observed in our cryoEM structure stabilizes the dimeric form. We have clarified this point in the revised results section.

Mg²⁺ vs. Mn²⁺ in Nuclease Assays:

In Figure 1d and most other nuclease assays, the authors use Mg²⁺ as the catalytic ion. However, Supplementary Figure 2c evaluates the influence of different metal ions on SLFN11 and SLFN14 nuclease activity. These data indicate that both Mg²⁺ and Mn²⁺ support comparable nuclease activity for SLFN14. In the case of SLFN11, Mn²⁺ leads to higher nuclease activity against tRNA^{Tser}. The data suggests that SLFN11 also exhibits nuclease activity with Mg²⁺ at high enzyme concentrations, but its enzymatic activity is higher in the presence of Mn²⁺. The authors should perform the ribosomal RNA cleavage assay additionally with Mn²⁺ and compare the results to those with Mg²⁺. The revised manuscript now includes Supplementary Figure 3, which characterizes SLFN14 ribosomal RNA cleavage activity in the presence of Mg²⁺ and Mn²⁺. We show that reaction conditions supplemented with either Mg²⁺ or Mn²⁺ support comparable levels of SLFN14 nuclease activity. We revised our proposed model to indicate that SLFN14 drives Mg²⁺/Mn²⁺-dependent RNA cleavage (please see Figure 6b).

Cryo-EM Density and Active Site Analysis:

The cryo-EM density does not clearly indicate whether one or two metal ions are bound in the active site. There appears to be unmodeled space in the density, which could accommodate a second metal ion or RNA. Given the flexibility and quality of the cryo-EM map in this region, modeling the RNA substrate and the active site is challenging. However, the current data do not support the conclusions drawn in Figures 5C and 6, as well as in lines 610, 648, and beyond. Similarly, the claim that residue orientations (E206 and D249) differ from those in other nucleases, including SLFN11, is not well supported by the density maps and models (Supplementary Figure 7f). The resolution does not allow for such definitive conclusions.

In all three cryo-EM maps, the first RNA base (chain D C1 and chain C G1) does not fit well within the cryo-EM density. However, unmodeled density closer to the active site suggests potential differences in RNA binding. The authors should consider performing a masked refinement including only the N-terminal and RNA-binding regions, without imposing C2 symmetry, to account for possible differences between the two active sites. Thank you for this feedback and suggestion. Masked refinement fails to produce high-quality maps of the NTD and RNA cleft regions, however we reprocessed our data with

C1 symmetry and identified two distinct states (state 1 and state 2) using 3D classification. In the revised manuscript, we replaced the C2 symmetric map and model with the asymmetric reconstructions. The resolution of state 2 is insufficient for model building; however, the reconstruction of state 1 has a resolution of 2.73 Å. We now present an asymmetric model of SLFN14 bound to RNA. In our asymmetric map, the active site density of protomer A is better resolved compared to our previous C2-symmetric map, whereas the side chain density in protomer B remains poorly resolved. We now state in the revised manuscript “*The reconstruction of state 1 was more clearly resolved for the active site in protomer A (chain A) compared to protomer B (chain B). For this reason, structural analysis and figure generation of the SLFN14 nuclease site were performed using protomer A.*” In the asymmetric map, we can model the active site side chains of protomer A, two coordinated metal ions, and nearby nucleotides. The resolution of the RNA density does not allow us to discern the identity of the bases nor whether this substrate is definitively a pre- or post-cleavage state. To emphasize these limitations, we state in the results section that the RNA sequence is arbitrary and added the statement “*Although this structural arrangement could, in principle, represent a scissile phosphate positioned for RNA hydrolysis or a post-cleavage product, the electron density within this region of the cryoEM reconstruction is not of sufficient quality to conclusively determine the RNA cleavage state. Despite these limitations, the structural organization of the SLFN14 active site observed in protomer A of our cryoEM reconstruction is supported by recent tRNA-bound SLFN11 pre- and post-cleavage structures, demonstrating that the N-lobes directly associate with the T-arm of type-I and type-II tRNAs, positioning the acceptor stem at the neighboring catalytic C-lobe.*”

Magnesium Coordination and Catalytic Mechanism:

Typically, Mg²⁺ is octahedrally coordinated, but in the SLFN14 structure, only three coordination sites are observed within a reasonable distance (2–4 Å). Given the missing model, additional unmodeled density in proximity, and the results in Supplementary Figure 2, the discussion on the Mg²⁺-dependent catalytic mechanism should be more cautious.

We agree. To address this important point, we reprocessed our cryoEM data without imposing C2 symmetry. The new asymmetric reconstruction of state 1 reveals improved density within the active site of protomer A. The density describing the active site side chains are improved, allowing for a comparative analysis between protomer A and other SLFN family members. We can also position two metals and an adjacent nucleotide, which along with well-resolved main chain and side chain features, more closely adopt the expected geometry for magnesium ion coordination. Alternatively, the active site of protomer B is poorly resolved and could not be used for analysis of the active site. The revised text now includes the statement “*The reconstruction of state 1 was more clearly resolved for the active site in protomer A (chain A) compared to protomer B (chain B). For this reason, structural analysis and figure generation of the SLFN14 nuclease active site were performed using protomer A.*”

Minor Comments:

Line 630: "RNA catalysis" should be corrected to "RNA cleavage."

This correction has been made to the revised text.

Supplementary Figures 1 and 2: The resolution in the PDF file is low and should be improved.

The corresponding figures were regenerated to increase the resolution of the images.

The manuscript has been substantially improved compared to the initial submission. However, several points should be either better discussed or validated using additional techniques.

We thank the reviewer for their constructive feedback, which has improved the revised manuscript.

Reviewer #2 (Remarks to the Author):

In their manuscript, Van Riper et al. report a functional and structural characterization of human SLFN14. Importantly, this manuscript is less a revision of the previous version submitted to NC but rather a new manuscript. Their conclusions are now based on an entirely new set of structural and functional data, leading to substantially different conclusions. Most importantly, their new data suggest that SLFN14 is not an anticodon-specific endonuclease, but rather exhibits specificity for the acceptor stem, very similar to other SLFN proteins. This new manuscript represents a significant improvement over the first version. The new enzymatic assays are well executed and now convincingly demonstrate the acceptor-stem specificity of SLFN14. Notably, they show that this specificity occurs only on native, presumably mature, tRNAs, whereas SLFN14 shows promiscuous cleavage on unmodified synthetic tRNAs. Unfortunately, the potential contribution of RNA modifications to this change in specificity remains unclear. One of the main issues raised previously by all three reviewers was the comparatively low resolution of the cryo-EM reconstruction. The authors now report a significantly improved cryo-EM reconstruction of SLFN14 to 2.5 Å, which even appears to have RNA bound in the active site cleft. Overall, this manuscript has significantly improved in data quality and clarity and will provide an important contribution to the field.

We thank the reviewer for their positive comments towards our revised manuscript. Please see below a point-by-point response to their constructive comments.

The authors should address the following points:

1. The map shows significant signs of orientation bias and oversharpening, especially in the NTDs of both subunits (e.g., bottom panel in Fig. 3a). This is also suggested by the angular distribution plots in Suppl. Fig. 6. Given that the interpretation on this region is critical for the main conclusions of this manuscript (e.g. including an inward-outward motion of the NTDs), it would be important either to solve the problem by collecting additional tilted data or rebalancing orientations, or at least to provide quantitative estimates for the orientation bias and include them into the discussion.

To address this comment along with the reviewer's second point described below, we reprocessed our cryoEM data to improve the quality of our 3D reconstructions. In the revised manuscript, we now report two distinct asymmetric reconstructions (state 1 and state 2) at a resolution of 2.73 Å and 3.11 Å, respectively. We also implement

alternative map sharpening strategies to minimize oversharpening (DeepEMhancer, Sanchez-Garcia et al. (2021) Commun. Biol.). While the 3D reconstruction of state 2 was insufficient for model building, we now present a revised model of SLFN14 bound to RNA using the asymmetric state 1 reconstruction. With significantly less map distortions and enhanced density within the active site of protomer A, we now report an updated model of RNA bound by juxtaposed N-lobes to align RNA towards one of its neighboring catalytic C-lobes (please see Figure 3). We have added additional statements throughout the results and methods sections to clearly describe the limitations of this study due to ambiguous features within the RNA density and active site density of protomer B. For example, we added the statement “*The reconstruction of state 1 was more clearly resolved for the active site in protomer A (chain A) compared to protomer B (chain B). For this reason, structural analysis and figure generation of the SLFN14 nuclease active site were performed using protomer A.*”

2. On p. 19 the authors state that “We did not observe asymmetric features and, therefore, applied C2 symmetry for the final consensus 3D reconstruction...”. This is problematic, since the authors report an RNA bound structure of SLFN14, which, necessarily, introduces asymmetry into the active site. Thus, with RNA bound, the two orientations of the dimer cannot be identical, and using C2 symmetry necessarily introduces artifacts into the RNA density. Maybe the authors could perform masked refinement focussed on the NTDs to solve the symmetry issue? This may also improve the overall quality of the local reconstruction in this region.

We agree. Unfortunately, masked refinement did not improve the local reconstruction of the RNA. Instead, we reprocessed our cryoEM data with C1 symmetry and identified two distinct states (state 1 and state 2). In the revised manuscript, we replaced the C2 symmetric map and model with an asymmetric model built from our new state 1 C1 map. In this new reconstruction, the active site density of protomer A is better resolved compared to our previous C2 map. We can confidently model the active site side chains of protomer A along with two coordinated metal ions and nearby nucleotides. We now report a dimeric SLFN14 model with a 3'-overhang RNA molecule (please see Figure 3c). Additional density is present near the active sites of protomer A and protomer B; however, due to its poor resolution, we left these regions of the map unmodeled. We also added statements describing the density quality of the C1 reconstruction. For example, we added the statement “*Although this structural arrangement could, in principle, represent a scissile phosphate positioned for RNA hydrolysis or a post-cleavage product, the electron density within this region of the cryoEM reconstruction is not of sufficient quality to conclusively determine the RNA cleavage state. Despite these limitations, the structural organization of the SLFN14 active site observed in protomer A of our cryoEM reconstruction is supported by recent tRNA-bound SLFN11 pre- and post-cleavage structures, demonstrating that the N-lobes directly associate with the T-arm of type-I and type-II tRNAs, positioning the acceptor stem at the neighboring catalytic C-lobe.*”

3. On p. 20 the authors discuss the possible source/identity of the bound RNA and suggest it may be co-purified native nucleic acid. This could be tested by looking at the dataset that was collected without the addition of tRNA during sample preparation.

Thank you for this suggestion. We revisited our cryoEM data prepared without tRNA supplementation. Unfortunately, the data was insufficient (preferred orientation) to draw a confident conclusion. Instead, we now include mass photometry data of the SLFN14 sample (please see Supplementary Figure 2). This new analysis demonstrates that in the absence of tRNA supplementation, the molecular mass of the SLFN14 protein is consistently larger than expected for apo protein (calculated dimer 232 ± 5.89 kDa vs. theoretical dimer 215 kDa). To promote the release of any potentially co-purified nucleic acid, we subjected the SLFN14 sample to high salt conditions. At 500 mM NaCl, the mass of the SLFN14 dimer (212 ± 3.21 kDa) aligns well with the expected mass of an apo dimer. One possible explanation for this calculated size discrepancy (20 ± 6.71 kDa) is the co-purification of cellular-sourced nucleic acid. This conclusion is supported by a characteristic absorbance of the SLFN14 protein sample at a wavelength of 260 nm. However, current attempts have failed to identify the bound nucleic acid.

4. On p. 22 the authors state that “The SLFN14•RNA structure reveals a sequence-independent RNA binding surface along its N-lobes”. Although sequence-independent electrostatic interactions indeed seem to make up much of this interface, the density for protein side chains in the interface and in particular the density for the RNA do not seem of sufficient resolution/quality to allow such an assertion.

To improve the resolution of the RNA and side chains along the RNA interface, we have reprocessed the cryoEM data (please see our point 2 response for further details). The new asymmetric reconstruction reveals improved density at the RNA interface. Many of the residues within the RNA interface are now well-resolved (e.g. protomer A residues K38, Q78, D79, T82, R127, S137, R133, E206, E211, K213, Y231). In the revised text, we limit our discussion to the potential role of individual residues with sufficient density in the asymmetric 3D reconstruction. Furthermore, we also included a statement of caution when describing regions with conformational heterogeneity. For example, when describing the disease hotspot, we now state “*While the electron density describing the side chains of basic residues K218, K219, and R223 are poorly defined in our cryoEM reconstruction, the well-resolved main chain places them at the transition point between an amphipathic α -helix and a partially ordered loop.*”

5. On p. 26 the authors discuss the “Structural homology with prokaryotic restriction endonucleases”. Do the authors claim this to be an actual “homology” between SLFN proteins and restriction endonucleases (i.e., due to descent from a common ancestor)? In this case they should demonstrate that this homology extends beyond the active site to the overall fold of both protein families. Alternatively, the authors should instead use the term “analogy” to describe convergent similarities in the active site architectures.

Thank you for raising this point. We have corrected the text by replacing structural homology with the term analogy.

Reviewer #3 (Remarks to the Author):

Van Riper et al., substantially revamped their study on human SLFN14, and now reports a 2.5 angstrom resolution cryo-EM structure of SLFN14 bound to a double-stranded RNA helix, possibly as part of a bound tRNA acceptor stem. The authors have also re-

examined the tRNA cleavage sites by SLFN14 and SLFN11, and found that SLFN14 cleavage sites can depend on tRNA modifications. These major changes, in particular the dramatic resolution improvement of their structures, have greatly improved the study. I believe the new conclusions reached in the revamped manuscript are now well supported by their experimental structure and associated data. I now support the publication of this work.

We thank the reviewer for their strong support of our revised manuscript.

NCOMMS-23-46926B Rebuttal

Reviewer comments (black) and our response (blue).

REVIEWERS' COMMENTS

Reviewer #1 (Remarks to the Author):

The authors have made significant improvements in response to reviewer feedback over the course of the revisions. The manuscript is now scientifically sound and clearly written. All previously raised concerns have been satisfactorily addressed.

We thank reviewer 1 for their strong support of our work and their valuable feedback.

Reviewer #2 (Remarks to the Author):

The authors have addressed all my concerns.

One minor issue is that the authors refer to the a Coulomb potential map as "electron density", which should be corrected.

We agree. We replaced "electron density" with "cryo-EM reconstruction."

Apart from that, I can recommend the manuscript for publication in Nature Communications

We thank reviewer 2 for their insightful feedback and support for publication.